# Surface-bound reactive oxygen species generating nanozymes for selective antibacterial action

Feng Gao [1,2,3,4], Tianyi Shao[1,3,4], Yunpeng Yu [1,2,3], Yujie Xiong [1,3✉] & Lihua Yang [1,2,3✉]

Acting by producing reactive oxygen species (ROS) in situ, nanozymes are promising as antimicrobials. ROS' intrinsic inability to distinguish bacteria from mammalian cells, however, deprives nanozymes of the selectivity necessary for an ideal antimicrobial. Here we report that nanozymes that generate surface-bound ROS selectively kill bacteria over mammalian cells. This result is robust across three distinct nanozymes that universally generate surface-bound ROS, with an oxidase-like silver-palladium bimetallic alloy nanocage, $AgPd_{0.38}$, being the lead model. The selectivity is attributable to both the surface-bound nature of ROS these nanozymes generate and an unexpected antidote role of endocytosis. Though surface-bound, the ROS on $AgPd_{0.38}$ efficiently eliminated antibiotic-resistant bacteria and effectively delayed the onset of bacterial resistance emergence. When used as coating additives, $AgPd_{0.38}$ enabled an inert substrate to inhibit biofilm formation and suppress infection-related immune responses in mouse models. This work opens an avenue toward biocompatible nanozymes and may have implication in our fight against antimicrobial resistance.

[1] Hefei National Laboratory for Physical Sciences at the Microscale, University of Science and Technology of China, Hefei, Anhui, China. [2] CAS Key Laboratory of Soft Matter Chemistry, University of Science and Technology of China, Hefei, Anhui, China. [3] School of Chemistry and Materials Science, University of Science and Technology of China, Hefei, Anhui, China. [4] These authors contributed equally: Feng Gao, Tianyi Shao. ✉email: yjxiong@ustc.edu.cn; lhyang@ustc.edu.cn

Antimicrobial resistance (AMR) poses a global threat to public health[1]. Depending on whether the resistance is genetically encoded or purely phenotypic, AMR can be inherited or non-inherited[2]. Inherited AMR results from mutations in existing genes or acquisition of nonnative resistance-encoding genes (via horizontal gene transfer). For example, methicillin-resistant *Staphylococcus aureus* (MRSA), one of the 12 antibiotic-resistant bacteria and bacterial families that pose the greatest threat to human health[3], acquires the resistance usually by the acquisition of *mecA*, a nonnative gene that encodes a penicillin-binding protein (PBP2a) with significantly lower affinity for β-lactam antibiotics[4]. Non-inherited AMR, on the other hand, denotes a phenomenon that bacteria, though genetically homogenous and inherently susceptible to antibiotics, become transiently refractory to the action of these drugs[2]. A notorious form of non-inherited AMR is a biofilm, in which bacterial cells are protected from antibiotics by the matrix of extracellular polymeric substances and/or the insufficient metabolism and replication characteristic of their living state therein[5,6]. Both the inherited and non-inherited AMR could cause treatment failure, and the latter could promote the generation and ascent of the former in treated patients[2]. In 2014, the burden of deaths from AMR was estimated to be at a minimum of 700,000 lives every year[1] and, unless action is taken, could balloon to 10 million lives each year by 2050, at a cumulative cost to global economic output of 100 trillion USD[1]. Accompanying this AMR crisis is, unfortunately, a failure in new antimicrobial discovery. Both the number of novel antibiotic types[7] and that of new antibiotics[8] approved by the regulatory agencies have been decreasing ever since the 1980s. Clearly, novel antibacterial agents are in urgent need[1].

Recently, nanozymes have been proposed as promising antimicrobials[9], thanks to their ability to kill bacteria with reactive oxygen species (ROS) they produce spontaneously in situ[10–18]. Because ROS simultaneously oxidize diverse cellular substances (e.g., nucleic acids, proteins, and lipids) crucial for proper cell function[19], nanozymes eliminate drug-resistant bacteria[18] and may delay the onset of bacterial resistance, suggesting potential for addressing genetically encoded AMR. Moreover, some nanozymes (e.g., $V_2O_5$ nanoparticles[10]) when used as coating additives enable the substrate surfaces to inhibit biofilm formation[10], suggesting potential for addressing phenotypic AMR. Despite these advantages, nanozymes are in general toxic to both bacteria and mammalian cells (Supplementary Table 1), indicative of a lack of the selectivity necessary for ideal antimicrobials, due to ROS' intrinsic inability to distinguish bacteria from mammalian cells. Clearly, making nanozymes preferentially active against bacteria over mammalian cells is very attractive; yet, how to achieve this remains a significant challenge.

Toxic ROS have very short lifetimes[20,21] and consequently limited effective radii of action (<200 nm)[20,22]. On the other hand, bacteria cannot engulf extracellular nanoparticles—those larger than certain threshold sizes (e.g., ~4 nm for gold nanoparticles (Supplementary Fig. 1), ~40 nm for silver nanoparticles[23]) remain extracellular unless bacterial cell walls are impaired[18]—whereas mammalian cells readily internalize nanoparticles via endocytosis[24], thereby trapping the particles within endocytotic vesicles, which can be abundant in the cytosol and whose disruption does not necessarily lead to cell death. Combined, these facts suggest that a nanozyme may preferentially kill bacteria over mammalian cells if the ROS it generates is surface-bound.

Here, we show that nanozymes that generate surface-bound ROS selectively kill bacteria over mammalian cells. The selectivity is attributable to the surface-bound nature of ROS these nanozymes generate and an unexpected antidote role of endocytosis, a cellular process that is common to mammalian cells but absent in bacteria. Though generating surface-bound ROS, these nanozymes have the potentials for efficiently eliminating antibiotic-resistant bacteria while delaying bacterial resistance emergence and for effectively inhibiting biofilm formation without causing damages to the host when used as coating additives. Such biocompatible nanozymes may have implications in our fight against both genetically encoded and phenotypical AMR.

## Results and discussion

Oxidase-like metal and alloy nanoparticles in principle catalyze the production of metal surface-bound ROS[25]. Palladium (Pd) nanocrystals are oxidase-like[25,26] but prohibitively expensive. In contrast, silver (Ag) nanoparticles, though affordable, in principle lack oxidase-like activities[25]. Compared to pure metals, alloys have more symmetrical unique adsorption sites for catalyzing the production of surface-bound ROS and may hence allow greater reactivity and more flexible design[25]. Therefore, we prepared porous AgPd bimetallic alloy nanocages (Fig. 1a) (via a galvanic replacement method using Ag nanocubes (AgNC) (Supplementary Fig. 2) as the sacrificial templates[27]) that are similar in size (60–75 nm under TEM, 125–145 nm in water) and surface zeta potential (ζ-potential, −17.1 to −3.8 mV) but differ in Pd molar content (0.08–0.50) (Fig. 1a and Supplementary Fig. 3). (For the same nanocage, its hydrodynamic diameter is larger than its size under TEM, because the former, but not the latter, includes the contribution by stealth coating materials, which in this case is poly(vinylpyrrolidone) (PVP, MW ~55,000) introduced in the preparation process. In fact, owing to the presence of PVP, these AgPd nanocages exhibited good colloidal stability in phosphate buffer solution (PBS), as indicated by the negligible changes in their hydrodynamic diameters over a span of 72 h (Supplementary Fig. 3d)).

We then screened these AgPd nanocages for oxidase-like activities. Ascorbic acid (AA) is an antioxidant whose strong absorption at 266 nm disappears upon oxidation[28]. Using AA as the probe for diverse ROS, we found that, after 3-h co-incubation in phosphate-buffered saline (PBS), AgPd nanocages reduced AA's absorbance at 266 nm to a varying extent depending on Pd content (Fig. 1b and Supplementary Fig. 4), indicative of nanocage-mediated production of ROS at differing efficiencies. Similar results were observed when the co-incubation time was shortened to 1 h (Supplementary Fig. 5). Notably, increasing the Pd content in AgPd nanocage did not necessarily lead to more ROS production; instead, there existed a non-monotonic relationship between ROS production versus Pd content, with $AgPd_{0.08}$ and $AgPd_{0.38}$ being the least and most efficient analogs, respectively (Fig. 1b). This non-monotonic relationship may arise because our nanocages—prepared via a galvanic replacement method[27]—may be partially alloyed and the strain due to lattice mismatch at the grain interfaces (AgPd alloy versus Ag versus Pd) in a nanocage, as exemplified with $AgPd_{0.38}$ (Supplementary Fig. 6), may control the catalyzing activity, as observed previously with the partially de-alloyed fuel cell catalysts[29]. It is noteworthy that AgPd nanocages produce ROS without requiring any externally applied stimulus. Moreover, using $AgPd_{0.38}$ as a representative for our AgPd nanocages, we revealed that the observed ROS production was nanocage dose-dependent (Supplementary Fig. 7) yet barely affected by a change in pH (Supplementary Fig. 8), temperature (Supplementary Figs. 9 and 10), or buffer agent (Supplementary Fig. 11). Therefore, incubation at 37 °C in PBS at pH ~ 7.4 was used for all following experiments unless specified otherwise.

Natural oxidases use oxygen as the substrate. In principle, oxidase-like metal and alloy nanomaterials catalyze the dissociation of $O_2$ into oxygen adatoms[25]. To examine whether $O_2$ is the substrate for our $AgPd_{0.38}$ nanocages, we used $AgPd_{0.38}$ as the

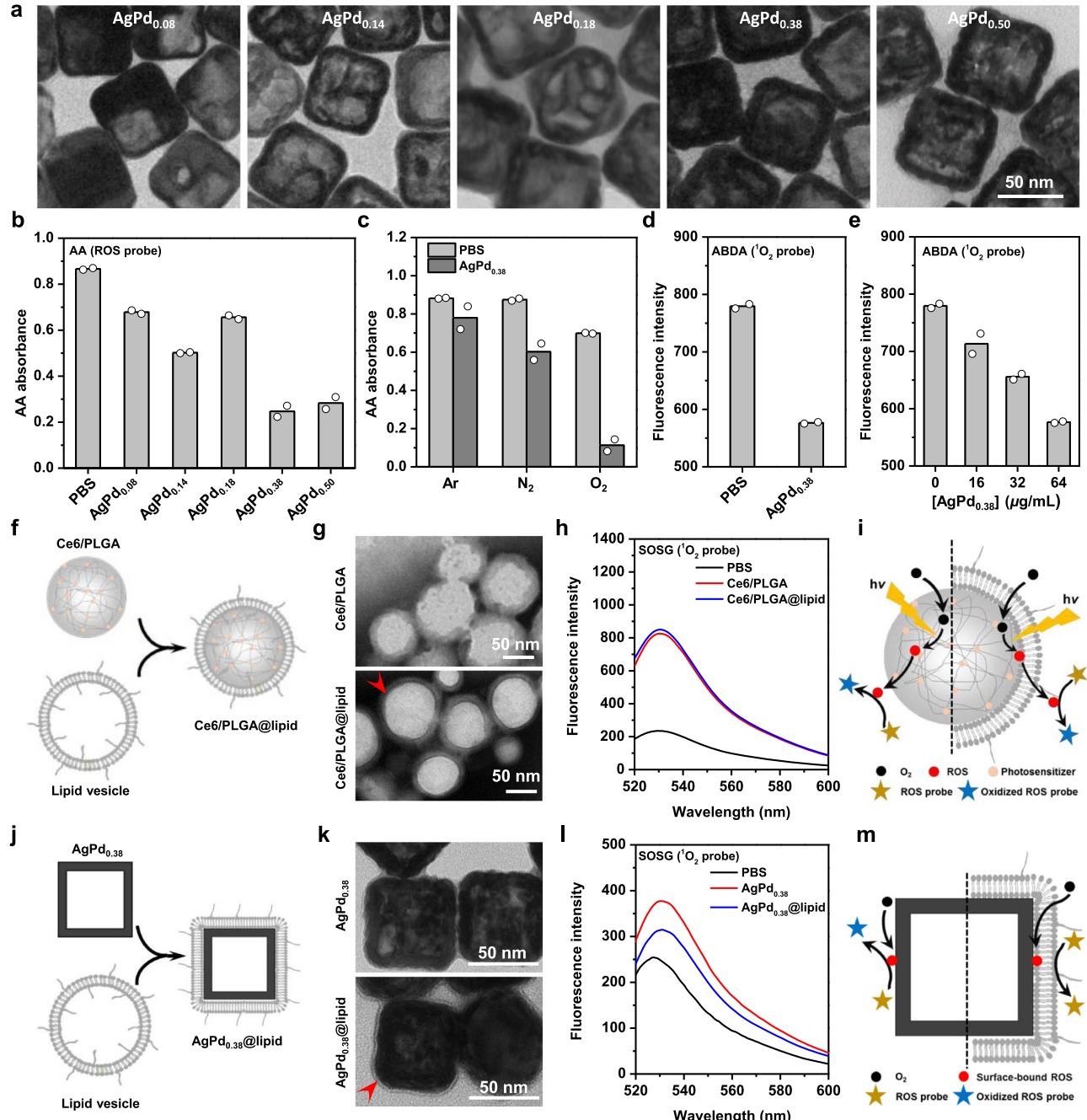

**Fig. 1 Oxidase-like AgPd$_{0.38}$ generates surface-bound ROS. a** Transmission electron microscopy (TEM) images of AgPd nanocages with different Pd content. **b** Absorbance ($\lambda = 266$ nm) of ascorbic acid (AA) treated (for 3 h) with a nanocage (8 µg/mL in PBS) or PBS. **c** Absorbance ($\lambda = 266$ nm) of AA treated with AgPd$_{0.38}$ (8 µg/mL) in the presence of O$_2$, N$_2$, or Ar. Controls are AA-treated similarly but with PBS. **d** Fluorescence intensity ($\lambda_{em} = 433$ nm) of ABDA treated with AgPd$_{0.38}$ (64 µg/mL) or PBS. **e** Fluorescence intensity ($\lambda_{em} = 433$ nm) of ABDA treated with AgPd$_{0.38}$ at different concentrations. **f** Schematic illustration on the preparation of Ce6/PLGA@lipid, which was done by coating a Chlorin e6 (Ce6)-preloaded PLGA (poly(lactic-co-glycolic acid)) nanoparticle (i.e., Ce6/PLGA) with a PEGylated lipid bilayer. **g** TEM images of Ce6/PLGA and Ce6/PLGA@lipid. Red arrow indicates the lipid bilayer coating. **h** Fluorescence emission spectra of SOSG treated with Ce6/PLGA@lipid or Ce6/PLGA (5 µg/mL in Ce6 dose) upon irradiation with a solar simulator (at 0.1 W/m$^2$, 5 min). Control is SOSG-treated similarly but with PBS. **i** Schematic illustration on the inability of the lipid bilayer coating in Ce6/PLGA@lipid to affect the outward translocation of free ROS generated by PLGA-encapsulated Ce6 upon light irradiation, thereby leading to unaffected oxidation of ROS probes in the bulk solution compared to Ce6/PLGA. **j** Schematic illustration on the preparation of AgPd$_{0.38}$@lipid, done by coating AgPd$_{0.38}$ with a PEGylated lipid bilayer. **k** TEM images of AgPd$_{0.38}$ and AgPd$_{0.38}$@lipid. Red arrow indicates the lipid bilayer coating. **l** Fluorescence emission spectra of SOSG treated with AgPd$_{0.38}$@lipid or AgPd$_{0.38}$ (64 µg/mL, for 3 h), with that of SOSG-treated similarly but with PBS included as a control. **m** Schematic illustration on the strikingly suppressed oxidation of ROS probes in the bulk solution by AgPd$_{0.38}$@lipid, as compared to that by AgPd$_{0.38}$, indicative of effective separation of the ROS generated by AgPd$_{0.38}$ from the ROS probes in the bulk solution due to the presence of the lipid bilayer coating, suggesting that the ROS on AgPd$_{0.38}$ is surface-bound. Source data are provided in the Source Data file.

representative and tested how $O_2$ removal affects its ROS production. Briefly, we removed dissolved $O_2$ via nitrogen ($N_2$) purging (for 30 min) from both the nanocage dispersion and the AA solution prior to their mixing, incubated the resulting mixture for 3 h, and then monitored the absorption spectra of AA therein. We found that $AgPd_{0.38}$, though capable of significantly reducing AA's absorbance in the presence of $O_2$, barely impacted that in the absence of $O_2$ (i.e., in the presence of $N_2$) (Fig. 1c and Supplementary Fig. 12a), indicative of strikingly suppressed ROS production due to $O_2$ removal. Similar results were observed by removing $O_2$ via Argon (Ar) purging (Fig. 1c and Supplementary Fig. 12b). Clearly, to produce ROS, $AgPd_{0.38}$ uses $O_2$, rather than $N_2$ or Ar, as the substrate. Moreover, after 3- or 18-h incubation, $AgPd_{0.38}$ exhibited negligible release of Ag and Pd (Supplementary Fig. 13), indicative of an undetectable change in nanocage composition, a basic requirement which nanozymes should meet. Collectively, these results suggest $AgPd_{0.38}$ be oxidase-like.

Some ROS species are highly reactive while others poor oxidants[30]. Being highly reactive, singlet oxygen ($^1O_2$) and hydroxyl radical (•OH)[19] are the ROS species that most nanozymes are developed to produce[10–14]. Similar to $^1O_2$ and •OH, oxygen adatoms—the ROS which oxidase-like metal and alloy nanoparticles in principle produce[25]—are highly reactive and, in particular, those generated on Pd nanocrystals resemble $^1O_2$ in chemical reactivity[31]. AgPd bimetallic alloy nanocages contain Pd in addition to Ag. A natural question that emerges next is: Does the ROS generated on our AgPd nanocages resemble $^1O_2$ in chemical reactivity as well? To address this question, we used $AgPd_{0.38}$ as the representative for our ROS-producing nanocages and examined whether the ROS it produces reacts with $^1O_2$ probes. 9,10-anthracenediyl-bis(methylene)dimalonic acid (ABDA) is a fluorescent molecule but upon capturing $^1O_2$ becomes non-fluorescent[32]. Using ABDA as the $^1O_2$ probe, we found that, after 3-h co-incubation with ABDA, $AgPd_{0.38}$ (at 64 μg/mL) reduced ABDA's fluorescence relatively by ~26% (Fig. 1d) and its ability to do so was nanocage dose-dependent (Fig. 1e and Supplementary Fig. 14), indicating production of $^1O_2$-like ROS by $AgPd_{0.38}$. ABDA loses its fluorescence upon oxidation by $^1O_2$. To exclude the possibility that the observed reduction in ABDA's fluorescence by $AgPd_{0.38}$ may arise because of ABDA's unexpected bleach, we carried out similar assays but replacing ABDA with singlet oxygen sensor green (SOSG), which is weakly blue fluorescent but upon capturing $^1O_2$ becomes brightly green fluorescent[31], and observed nanocage dose-dependent enhancement in SOSG's green fluorescence (Supplementary Fig. 15). Taken together, these observations suggest that the ROS produced by $AgPd_{0.38}$ resemble $^1O_2$ in chemical reactivity. Of note, $AgPd_{0.38}$ does not generate ROS that resembles •OH in chemical reactivity, as $AgPd_{0.38}$ failed to impact the fluorescence of p-phthalic acid (PTA) (Supplementary Fig. 16), which is virtually non-fluorescent but upon capturing •OH becomes brightly fluorescent ($\lambda_{em} = 400$ nm)[33]. Taken together, these results suggest that $AgPd_{0.38}$ produces ROS that resembles $^1O_2$, but not •OH, in chemical reactivity.

Next, we set to examine whether the ROS generated on $AgPd_{0.38}$ are surface-bound, especially considering that the energy input ($\Delta E \sim 1.6$ eV)[34] necessary for converting a free $O_2$ into a free $^1O_2$ was absent under our assay conditions and that oxidase-like metal and alloy nanomaterials are calculated to produce oxygen adatoms[25]. Lipid bilayers have thicknesses that are normally shorter than the effective radii of action of ROS (20–200 nm)[20,22]. Moreover, a lipid bilayer is permeable to $O_2$ but impermeable to AA[35] or SOSG[36]. In addition, using Chlorin e6 (Ce6)-preloaded PLGA (poly(lactic-co-glycolic acid)) nanoparticle (i.e., Ce6/PLGA) as a model for free $^1O_2$-generating nanoparticles (Ce6 is an organic photosensitizer that generates free $^1O_2$ upon near-infrared light irradiation ($\lambda \sim 660$ nm)[37,38]), we observed that coating Ce6/PLGA with a lipid bilayer

(composed of DOPC: DSPE-PEG = 0.90: 0.10) (Fig. 1f, g) that is ~11 nm in thickness (Supplementary Fig. 17a) barely affected the particle's ability to oxidize SOSG in the bulk solution upon light irradiation (Fig. 1h), indicative of non-slowed translocation of free $^1O_2$ across the lipid bilayer and negligible consumption of $^1O_2$ by the lipid bilayer coating despite its significant content of unsaturated lipid (DOPC at 90% weight percentage), suggesting that our lipid bilayer coating is permeable to free $^1O_2$ (Fig. 1i). To our surprise, coating $AgPd_{0.38}$ with the same lipid bilayer (Fig. 1j, k and Supplementary Fig. 18) significantly impaired the nanocage's ability to oxidize SOSG in the bulk solution (Fig. 1l). Consistently, the as-coated $AgPd_{0.38}$@lipid particle is significantly less efficient in oxidizing AA in the bulk solution than bare $AgPd_{0.38}$ (Supplementary Fig. 19). Clearly, the presence of this lipid bilayer coating imposed distinct effects on the ability of the ROS generated by $AgPd_{0.38}$ versus light-irradiated Ce6/PLGA to oxidize lipid bilayer-impermeable probes in the bulk solution. And the observed distinction must arise because the ROS generated by $AgPd_{0.38}$ are surface-bound (Fig. 1m), rather than freely floating as those by light-irradiated Ce6/PLGA.

We next examined whether AgPd nanocages preferentially kill bacteria over mammalian cells as expected. In vitro antibacterial assays (Fig. 2a and Supplementary Fig. 20) revealed that, as Pd content increases, our AgPd nanocages exhibited antibacterial profiles ranging from completely inactive (the minimum concentrations to kill 99.9% of inoculated bacterial cells, $MBC_{99.9}$, ≥128 μg/mL) to active against a wide spectrum of bacteria ($MBC_{99.9}$ of 4- 64 μg/mL), with $AgPd_{0.08}$ and $AgPd_{0.38}$ being the least and most potent, respectively; Staphylococcus aureus and Bacillus subtilis were used as the representatives for Gram-positive bacteria while Escherichia coli and Pseudomonas aeruginosa the representatives for Gram-negative bacteria. Of note, the relationship between $MBC_{99.9}$ value versus Pd content was not only non-monotonic (Fig. 2a) but also perfectly mirrored that between ROS production versus Pd content (Fig. 1b and Supplementary Fig. 21). Moreover, pre-mixing $AgPd_{0.38}$ with carotene, a scavenger of $^1O_2$[26], prior to its exposure to bacteria rendered $AgPd_{0.38}$ completely inactive (Fig. 2b and Supplementary Fig. 22). Collectively, these results suggest oxidase-like AgPd nanocages be antibacterial and identify $AgPd_{0.38}$ as the most potent analog.

ROS-generating nanozymes can circumvent bacterial resistance mechanisms[19], because ROS simultaneously disrupt diverse cellular substances (e.g., nucleic acids, proteins, and lipids) crucial for proper cell function, rather than targeting specific intracellular metabolic pathways as do antibiotics (if not all)[39]. Indeed, using MRSA as the representative for antibiotic-resistant bacteria (Fig. 2c and Supplementary Fig. 23), we found that, against MRSA, $AgPd_{0.38}$ exhibited an $MBC_{99.9}$ (16 μg/mL) nearly identical to that against the antibiotic-sensitive laboratory S. aureus strain, indicative of potency against antibiotic-resistant bacteria. Moreover, $AgPd_{0.38}$ delayed the onset of bacterial resistance following repeated treatment, according to serial bacterial inhibition assays[40], whose applicability was verified with gentamicin and levofloxacin (two antibiotics) (Supplementary Figs. 24–26). Against both E. coli and S. aureus, $AgPd_{0.38}$ exhibited unchanged minimum concentrations to inhibit the growth of 90% inoculated bacterial cells ($MIC_{90}$) throughout 12 treatment passages (Fig. 2d, e and Supplementary Fig. 27). In stark contrast, silver nanoparticle (AgNP, ~20 nm in size) (Supplementary Fig. 28), which cannot generate ROS through oxidase-like activity[25] or via a Fenton-like reaction[41,42] under our assay conditions, exhibited 8- and 16-fold increase in $MIC_{90}$ against S. aureus and E. coli, respectively (Fig. 2d, e and Supplementary Fig. 29), indicative of resistance emergence, consistent with a prior report[40]. The reason why we used AgNP (~20 nm in size), rather than the silver nanocube (AgNC) precursor

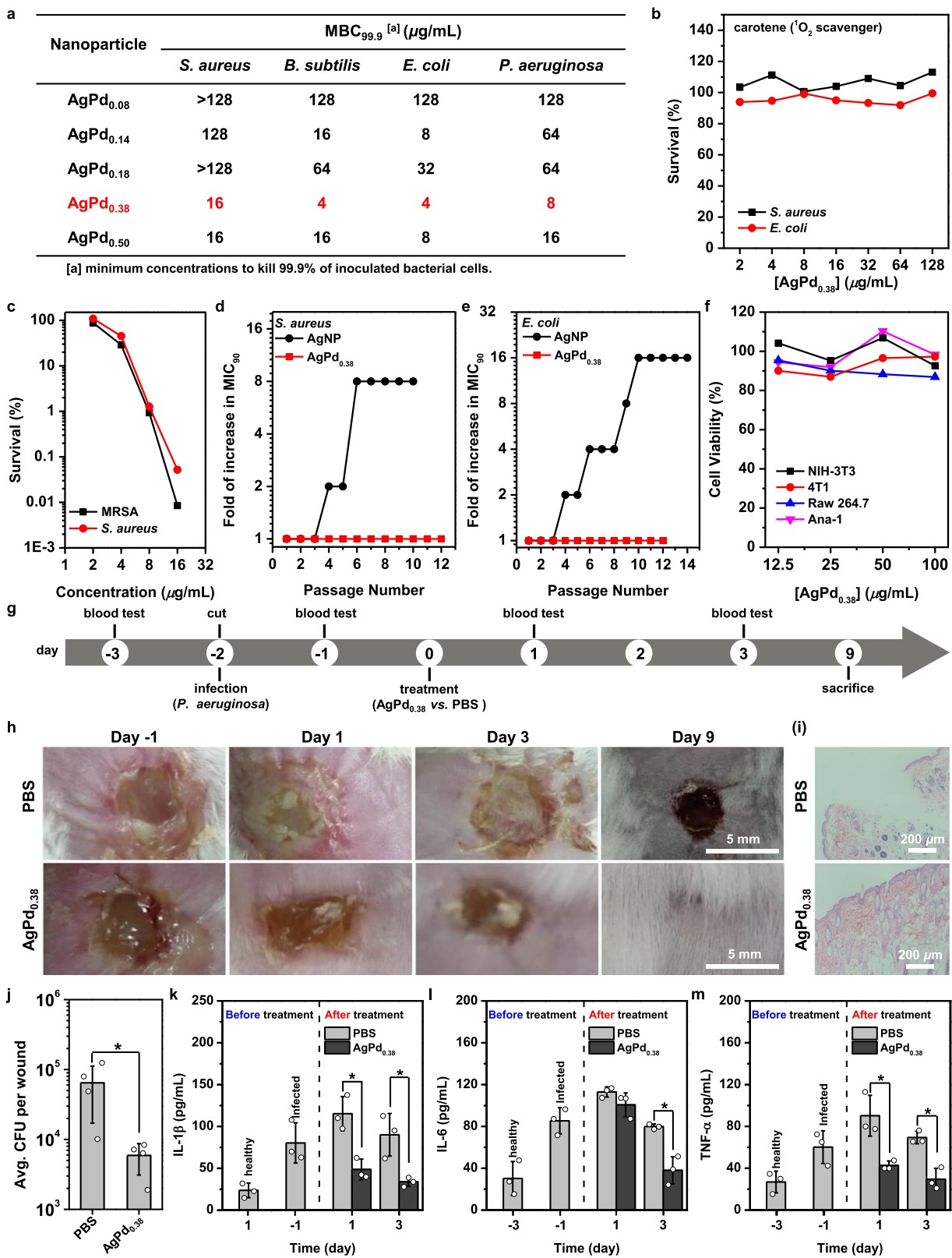

of our AgPd nanocages, is that our AgNC is inactive against bacteria (Supplementary Fig. 30). Clearly, AgPd$_{0.38}$ efficiently eliminated antibiotic-resistant bacteria and effectively delayed the onset of resistant emergence, owing to its ROS-producing ability.

We next examined whether AgPd$_{0.38}$ is cytotoxic to mammalian cells. Murine macrophage Raw 264.7, murine macrophage Ana-1, murine fibroblast NIH-3T3, and murine breast cancer cell 4T1 were used as the representatives for mammalian cell-lines. Intriguingly, AgPd$_{0.38}$ failed to impact the cell viability even at 100 μg/mL in cell counting kit-8 (CCK-8) assays (Fig. 2f and Supplementary Fig. 31) and this was the case with all four tested cell-lines, indicative of lack of in vitro cytotoxicity. Combined

**Fig. 2 AgPd$_{0.38}$ preferentially kills bacteria over mammalian cells. a–f** In vitro antibacterial assays and cell viability studies. **a** In vitro antibacterial potentials of AgPd nanocages, as indicated by MBC$_{99.9}$, the minimum nanocage concentration to kill 99.9% of inoculated bacterial cells. *Staphylococcus aureus* (*S. aureus*) and *Bacillus subtilis* (*B. subtilis*) were used as representatives for Gram-positive bacteria, while *Escherichia coli* (*E. coli*) and *Pseudomonas aeruginosa* (*P. aeruginosa*) were used as representatives for Gram-negative bacteria. **b** Plate-killing assays of AgPd$_{0.38}$ in the presence of carotene. **c** Plate-killing assays of AgPd$_{0.38}$ against methicillin-resistant *Staphylococcus aureus* (MRSA) or antibiotic-sensitive *S. aureus*. **d**, **e** Folds of increase in MIC$_{90}$ (the minimum concentration to inhibit the growth of 90% inoculated bacterial cells) of AgPd$_{0.38}$ through serial passages of growth inhibition assays against (**d**) *S. aureus* and (**e**) *E. coli*. Silver nanoparticle (AgNP, ~20 nm in size) was included as a model for nanoparticles that cannot generate ROS under similar conditions. **f** Cell viability assays of AgPd$_{0.38}$ to four mammalian cell-lines: murine macrophage Raw 264.7, macrophage Ana-1, murine embryo fibroblast NIH-3T3, and murine breast cancer 4T1. Each in vitro antibacterial or cell viability assay was carried out in triplicate and the reported results are averages of two independent trials. Plots on source data are provided as Supplementary Figures. **g–m** Experiments with mouse models. **g** Schedule of tests on the performance of AgPd$_{0.38}$ or PBS administrated topically to *P. aeruginosa*-infected wounds in mouse models. Six mice per treatment group. **h** Photographs of wounds from the two treatment groups throughout the observation window. **i** Microscopy images of hematoxylin and eosin (H&E) stained sections of wound tissues collected on day 9. **j** Average colony forming units (CFU) per wound for AgPd$_{0.38}$- or PBS-treated mice on day 9. Data points are reported as mean ± standard deviation (*n* = 4). **k–m** Serum levels of **k** IL-1β, **l** IL-6, and **m** TNF-α in blood samples collected on differing days from the mouse models. Data points are reported as mean ± standard deviation (*n* = 3). * Indicates *p* < 0.05, analyzed by two-sided Student's *t*-test. Source data are provided in a Source Data file.

with AgPd$_{0.38}$'s wide-spectrum antibacterial activity (MBC$_{99.9}$ of 4–16 μg/mL) (Fig. 2a), these results suggest that AgPd$_{0.38}$ preferentially kills bacteria over mammalian cells in vitro.

To evaluate whether AgPd$_{0.38}$ is preferentially active against bacteria over host cells under pathological conditions in vivo, we examined its performance in mouse models bearing wounds infected with *P. aeruginosa* (Fig. 2g), a most common causative bacteria of chronic wound infections[43], with the performance of PBS included as a negative control. Throughout the observation window, AgPd$_{0.38}$ significantly promoted wound healing (Fig. 2h), leading to restitution of normal tissue structure in the AgPd$_{0.38}$-treated wounds on day 9 when the observation was stopped (Fig. 2i). Moreover, the average colony-forming units (CFU) per wound in AgPd$_{0.38}$-treated mice was over one-magnitude lower than those in PBS-treated counterparts (Fig. 2j), indicative of significantly reduced bacterial burdens by AgPd$_{0.38}$. Interleukin-1β (IL-1β), interleukin-6 (IL-6), and tumor necrosis factor-α (TNF-α) are three pro-inflammatory cytokines vital to inflammatory disease initiation and progression[44–46]. At 24-h after *P. aeruginosa* infection (i.e., day −1), the serum levels of IL-1β, IL-6, and TNF-α were significantly higher than their respective healthy levels (i.e., day −3) (Fig. 2k–m), indicative of acute inflammatory response due to bacterial infection. In the absence of effective treatment (i.e., PBS), the serum levels of IL-1β, IL-6, and TNF-α continued to increase in the subsequent 48 h (i.e., day 1) and failed to drop back to normal even on day 3 despite the mouse innate immunity (Fig. 2k–m). Nevertheless, AgPd$_{0.38}$ treatment significantly reduced the serum levels of IL-1β and TNF-α and slowed the increase in that of IL-6 on day 1 (when the inflammatory response was most acute) and successfully lowered them close to healthy levels on day 3 (Fig. 2k–m), indicative effective suppression on the inflammatory response by AgPd$_{0.38}$, suggesting both effective antibacterial activity and lack of off-target toxicity. Besides, the topical application of AgPd$_{0.38}$ into wounds failed to damage the five major organs (which are liver, heart, kidney, spleen, and lung) (Supplementary Fig. 32), indicative of lack of nanotoxicity. Taken together, these results demonstrated that AgPd$_{0.38}$ retained its antibacterial activity and lack of off-target toxicity even under the complex pathological conditions in vivo.

A question that naturally emerges is: How AgPd$_{0.38}$ acquires the observed preferential killing of bacteria over mammalian cells? Bacteria cannot engulf nanoparticles. Relatively large in size (69.8 ± 8.9 nm under TEM, 137.9 ± 3.6 nm in water), AgPd$_{0.38}$ should not be able to enter bacteria unless bacterial cell walls are disrupted. In fact, AgNC—which is similar in size and morphology as AgPd$_{0.38}$ but completely inactive (Supplementary Fig. 30) and hence effectively an inactive phantom of AgPd$_{0.38}$—

remained extracellular and bacterial surface-associated in bacterial dispersions (Fig. 3a and Supplementary Fig. 33). Therefore, bacterial cell walls may be the target of AgPd$_{0.38}$. This was the case. Under scanning electron microscopy (SEM), AgPd$_{0.38}$-treated bacteria appeared perforated and even collapsed, whereas AgPd$_{0.08}$-treated counterparts remained smooth and intact as did the controls (i.e., treated similarly but with PBS) (Fig. 3b). For Gram-negative bacteria, outer membranes comprise an important part of their cell walls and also a significant barrier that protects them from commonly used antibiotics[47]. Using 1-*N*-phenylnaphthylamine (NPN) as a probe for outer membrane integrity, we found that AgPd$_{0.38}$, but not AgPd$_{0.08}$, significantly permeabilized the outer membranes of *E. coli* and *P. aeruginosa* (Supplementary Fig. 34). Moreover, bacterial live/dead viability assays showed that AgPd$_{0.38}$, but not AgPd$_{0.08}$, permeabilized bacterial cytoplasmic membranes (Supplementary Fig. 35). Apparently, AgPd$_{0.38}$ killed bacteria by disrupting their cell walls and cytoplasmic membranes.

In contrast to bacteria, mammalian cells readily internalize nanoparticles via endocytosis[24]. Using macrophage Raw 264.7 as the representative for mammalian cell-lines, we found that ~27% of the added AgPd$_{0.38}$ particles were uptaken by the AgPd$_{0.38}$-treated Raw 264.7 cells (Fig. 3c and Supplementary Fig. 36). It should be noted that the cellular uptake here included both surface-attached and intracellular AgPd$_{0.38}$. Decreasing the incubation temperature from 37 °C to 4 °C significantly suppressed the observed uptake of AgPd$_{0.38}$ (Fig. 3c), indicative of energy-dependence, suggesting endocytosis, rather than energy-independent ways (e.g., direct penetration through the cellular membrane), as the major mode in the cellular uptake of AgPd$_{0.38}$. Endocytosis occurs via diverse pathways (Fig. 3d)[24,48], among which phagocytosis is typically restricted to specialized cells (e.g., macrophages) that function to clear large pathogens and cell debris[24]. As neither macrophage Raw 264.7 nor fibroblast NIH-3T3 cells were appreciably eliminated by AgPd$_{0.38}$ (Fig. 2f), phagocytosis is thus unlikely a major endocytosis pathway for AgPd$_{0.38}$. Amiloride, sucrose, and methyl-β-cyclodextrin (M-β-CD) are inhibitors for micropinocytosis, clathrin-dependent endocytosis, and caveolae-dependent endocytosis, respectively[49] (Fig. 3d). For AgPd$_{0.38}$, sucrose, but neither amiloride nor M-β-CD, significantly inhibited its cellular uptake by Raw 264.7 cells (Fig. 3e and Supplementary Fig. 37), suggesting clatherin-mediated endocytosis as the major pathway by which AgPd$_{0.38}$ enters cells. Consistently, the cellular uptake of AgPd$_{0.38}$ was significantly suppressed by dynasore (Fig. 3e), an inhibitor for dynamin-related endocytosis[50] (including clatherin-mediated endocytosis, caveolin-dependent endocytosis, and the dynamin-

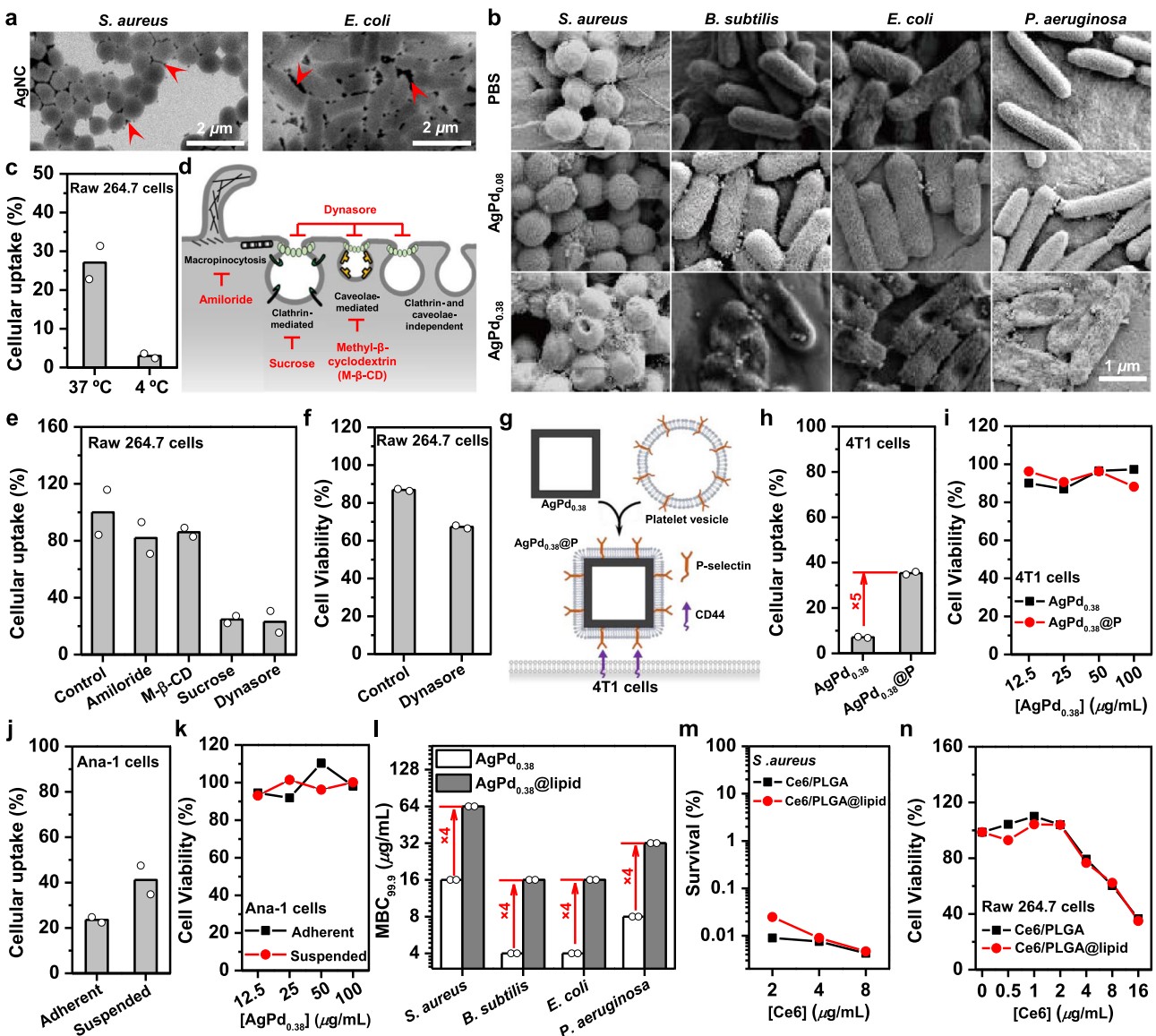

**Fig. 3 Origin of AgPd$_{0.38}$'s observed selectivity. a** Synchrotron soft X-ray microscopy images of silver nanocube (AgNC) in aqueous dispersions of *S. aureus* and *E. coli*. **b** Scanning electron microscopy (SEM) images of bacteria treated with AgPd$_{0.38}$, AgPd$_{0.08}$, and PBS. **c** Cellular uptake of AgPd$_{0.38}$ by Raw 264.7 cells at 4 and 37 °C. Cellular uptake here is defined as the relative mass ratio of AgPd$_{0.38}$ associated onto and internalized into cells to the total AgPd$_{0.38}$ added. **d** Schematic illustration on endocytosis pathways and their respective inhibitors. **e** Cellular uptake of AgPd$_{0.38}$ by Raw 264.7 cells in the presence of an inhibitor at 37 °C. Control denotes Raw 264.7 cells treated similarly but in the absence of any inhibitor. **f** Viability ratios of Raw 264.7 cells treated with AgPd$_{0.38}$ (100 μg/mL) in the presence of dynasore. Control denotes Raw 264.7 cells treated similarly but in the absence of dynasore. **g** Schematic illustration on the preparation of AgPd$_{0.38}$@P and the specific recognition between P-selectin naturally present on platelet membrane and CD44 receptors over-expressed on 4T1 cells. **h** Cellular uptake of AgPd$_{0.38}$@P and AgPd$_{0.38}$ by 4T1 cells. **i** Viability ratios of 4T1 cells after treatment with AgPd$_{0.38}$@P and AgPd$_{0.38}$. **j** Cellular uptake of AgPd$_{0.38}$ by adherent versus suspended Ana-1 cells. **k** Viability ratios of adherent versus suspended Ana-1 cells after treatment with AgPd$_{0.38}$. **l** MBC$_{99.9}$ values of AgPd$_{0.38}$@lipid versus AgPd$_{0.38}$. **m** Survival ratios of *S. aureus* cells after 10-min treatment with a photodynamic nanoparticle (Ce6/PLGA@lipid, or Ce6/PLGA) and then 5-min irradiation with a solar simulator (at 0.1 W/m²). **n** Viability ratios of Raw 264.7 cells after 4-h treatment with a photodynamic nanoparticle (Ce6/PLGA@lipid, or Ce6/PLGA) and then 5-min irradiation with a solar simulator (at 0.1 W/m²). Each in vitro antibacterial or cell viability assay was carried out in triplicate and the reported results are averages of two independent trials. Plots on source date are provided as Supplementary Figures. Source data are provided in the Source Data file.

mediated forms of clatherin-/caveolin-independent endocytosis) (Fig. 3d, e).

To understand whether endocytosis protected mammalian cells from AgPd$_{0.38}$, we examined how inhibiting AgPd$_{0.38}$'s endocytosis with dynasore affects AgPd$_{0.38}$'s cytotoxicity. To our surprise, we found that dynasore, though lacking intrinsic cytotoxicity (Supplementary Fig. 38), rendered AgPd$_{0.38}$ appreciably cytotoxic; in the presence of dynasore, AgPd$_{0.38}$ (100 μg/mL) eliminated ~35%

of treated Raw 264.7 cells (Fig. 3f and Supplementary Fig. 39), as compared to ~10% elimination in the absence of dynasore. Clearly, carrying surface-bound ROS, AgPd$_{0.38}$ could have been cytotoxic; nevertheless, endocytosis together with ROS' limited effective radii of action has unexpectedly protected mammalian cells from AgPd$_{0.38}$, likely by confining the reactivity of AgPd$_{0.38}$'s surface-bound ROS within endocytic vesicles. On the other hand, we assessed how promoting AgPd$_{0.38}$'s cellular uptake affects its

cytotoxicity. Note that the specific recognition between P-selectin naturally present on platelet membrane[51] and CD44 receptors over-expressed on cancer cells (e.g., murine breast cancer 4T1 cells)[51,52] promotes the uptake of platelet membrane-coated nanoparticles by cancer cells[21,53]. We hence coated $AgPd_{0.38}$ with a platelet membrane (Fig. 3g and Supplementary Fig. 40) and found that, though the presence of platelet membrane coating enhanced $AgPd_{0.38}$'s uptake by 4T1 cells relatively by ~4-fold (Fig. 3h and Supplementary Fig. 41), the as-coated $AgPd_{0.38}$ (i.e., $AgPd_{0.38}@P$) eliminated only <10% of 4T1 cells as did the bare $AgPd_{0.38}$ (Fig. 3i and Supplementary Fig. 42). Moreover, using murine macrophage Ana-1 as the representative for mammalian cell-lines, we found that, although $AgPd_{0.38}$'s uptake by suspended Ana-1 cells (~45%) was almost 2-fold of that by adherent Ana-1 cells (~25%) (Fig. 3j and Supplementary Fig. 43), this particle (up to 100 μg/mL) failed to eliminate 10% of either suspended or adherent Ana-1 cells (Fig. 3k and Supplementary Fig. 44). Collectively, these results indicate that inhibiting, but not promoting, $AgPd_{0.38}$'s uptake by mammalian cells makes this particle cytotoxic, suggesting that endocytosis unexpectedly protects mammalian cells from $AgPd_{0.38}$.

We next set to unveil whether the surface-bound nature of the ROS on $AgPd_{0.38}$ underlies its observed selectivity against bacteria over mammalian cells. Firstly, we coated $AgPd_{0.38}$ with a lipid bilayer (DOPC:DSPE-PEG = 0.90:0.10), a physical barrier which prevents the surface-bound ROS on $AgPd_{0.38}$ from getting into contact with cells (bacterial or mammalian) in the vicinity and compared the antibacterial activity of particle before and after the lipid bilayer coating. Intriguingly, the presence of this lipid bilayer coating significantly weakened $AgPd_{0.38}$'s antibacterial activity, as the as-coated particle (i.e., $AgPd_{0.38}@lipid$) exhibited $MBC_{99.9}$ values 4-fold higher than bare $AgPd_{0.38}$ (Fig. 3l and Supplementary Fig. 45) (The residual bactericidal activity of $AgPd_{0.38}@lipid$ may arise because of partial exposure of metal surface via oxidation of the lipid bilayer at sites in contact with the surface-bound ROS). To exclude the possibility that the observed weaker antibacterial activity of $AgPd_{0.38}@lipid$ than $AgPd_{0.38}$ arises because of lipid bilayer coating's consumption of free ROS or hindrance on ROS efflux, we used light-irradiated Ce6/PLGA as a model for free ROS-generating nanoparticles and found that coating Ce6/PLGA with a lipid bilayer failed to affect its antibacterial activity and cytotoxicity to mammalian cells upon light irradiation (Fig. 3m, n and Supplementary Fig. 46). Similar results were observed with verteporfin-preloaded PLGA-PEG nano-particle (i.e., Ver/PLGA) that upon light irradiation generates free $^1O_2$ as well (Supplementary Figs. 47 and 48). The distinct effects of lipid bilayer coating on the bioactivity of $AgPd_{0.38}$ versus free ROS-generating nanoparticles suggest that the observed selectivity of $AgPd_{0.38}$ must originate in the surface-bound nature of the ROS it generated (Fig. 1j–m).

We next set to understand whether preferentially killing of bacteria over mammalian cells is a global activity of surface-bound ROS-generating nanoparticles. Thermally reduced $TiO_2$ nanoparticle (i.e., $R-TiO_2$) was reported to spontaneously generate surface-stabilized superoxide radicals (through electron transfer from $Ti^{3+}$ defects in addition to oxygen vacancy sites formed during the thermal reduction to adsorbed $O_2$ on the $R-TiO_2$ surface) that are stable at ambient temperature (Fig. 4a)[54,55]. Indeed, $R-TiO_2$ nanoparticle in dark spontaneously generated superoxide radical whereas its unreduced precursor $TiO_2$ (Degussa P25) nanoparticle did not, according to fluorescence assays using Dihydroethidium (DHE) as the probe for superoxide radical (Supplementary Fig. 49). Using $R-TiO_2$ as a second model for surface-bound ROS-generating nanozymes, we found that, in the dark, $R-TiO_2$ (at 400 μg/mL) killed ~60% of inoculated $B.$ $subtilis$ and $E.$ $coli$ cells whereas its unreduced precursor $TiO_2$ failed to kill 10% of inoculated bacteria (Fig. 4b, c and

Supplementary Fig. 50). In contrast to its observed antibacterial activity, $R-TiO_2$ in the dark failed to eliminate ~10% of treated mammalian cells even up to 3200 μg/mL and this was the case with all three examined cell-lines (fibroblast NIH-3T3, macrophage Ana-1, and macrophage Raw 264.7) (Fig. 4d and Supplementary Fig. 51). Clearly, $R-TiO_2$, which generates surface-bound superoxide radical, preferentially eliminates bacteria over mammalian cells. In addition, $FeN_5$ SA/CNF, a single-atom nanozyme that contains carbon nanoframe-confined $FeN_5$ as the catalytically active centers and in principle generates oxygen adatoms, efficiently kills a wide spectrum of bacteria but barely impacts the viability of human colon mucosal epithelial NCM460 cells, thereby leading to effective wound disinfection without negatively affecting the skin tissue[56]. Inspiringly, our $AgPd_{0.38}$ nanocage, $FeN_5$ SA/CNF, and $R-TiO_2$ nanoparticles unanimously exhibited preferential killing of bacteria over mammalian cells, despite the distinction in their materials and in ROS types they generate. Such a strong similarity in bioactivity despite their distinction in materials encouraged us to propose that preferential killing of bacteria over mammalian cells is a global behavior for nanoparticles that spontaneously generate surface-bound ROS.

Medical devices (e.g., catheters, contact lenses) are widely used nowadays. Yet, their use is challenged significantly by the risk of device-related infections. Failed devices frequently reveal the formation of biofilms, which shelter the causative bacteria from the attack by host immune responses and antibiotic therapies. To protect a device surface from bacterial colonization, it is necessary to modify it with materials that repel bacterial attachment (i.e., antifouling) and/or kill bacteria (i.e., antibacterial). For example, pre-impregnating a substrate with antibiotics readily offers antibacterial potency; yet, the resulting surface fails when the pre-loaded antibiotics are depleted or when the causative bacteria are antibiotic-resistant and/or Gram-negative bacteria (the outer membrane of Gram-negative bacteria confers insensitivity to commonly used antibiotics[47]). Polydimethylsiloxane (PDMS) is a material frequently used to manufacture plastic biomedical devices[57]. We hence used a circular wafer of PDMS as the model for biomedical device surfaces, coated it successively and successfully with dopamine and $AgPd_{0.38}$ (at $11.23 \pm 1.40$ μg/cm$^2$) (Fig. 5a, b and Supplementary Figs. 52 and 53), and examined the antibacterial potential of $AgPd_{0.38}$ as surface coating additive. Using bovine serum albumin (BSA) as a model for proteins available in body fluids, we found that, although BSA readily adsorbed onto both the pristine PDMS and the intermediate dopamine-coated PDMS (i.e., PDMS/PDA), it exhibited negligible adsorption on PDMS/PDA/AgPd (Supplementary Fig. 54a), indicative of antifouling effects due to $AgPd_{0.38}$'s presence. After 3-h co-incubation with planktonic bacteria, PDMS/PDA/AgPd killed ≥85% of inoculated bacterial cells in the bulk solution (Supplementary Fig. 54b), despite that PDA/PDMS and bare PDMS were completely inactive, suggesting that $AgPd_{0.38}$ as surface coating additive enables an otherwise inactive substrate to kill planktonic bacteria. $P.$ $aeruginosa$ PAO1 is a Gram-negative bacterium that is intrinsically green fluorescent[58] and has a strong tendency to form biofilms[59] and hence a widely used representative for biofilm-causative bacteria. Intriguingly, after 5-day incubation with $P.$ $aeruginosa$ PAO1 in nutrient broth, PDMS/PDA/AgPd appeared to be free of attached bacteria under SEM (Fig. 5c, 2nd row), despite that both the pristine PDMS and the intermediate dopamine-coated PDMS (i.e., PDMS/PDA) exhibited significant amounts of surface-attached cells. Consistently, under confocal fluorescence microscopy (Fig. 5c, 3rd row), PDMS/PDA/AgPd surface remained dark in the green channel whereas both PDMS and PDMS/PDA surfaces were covered by brightly green fluorescent substances despite the absence of any staining process in our assays, indicative of colonization of

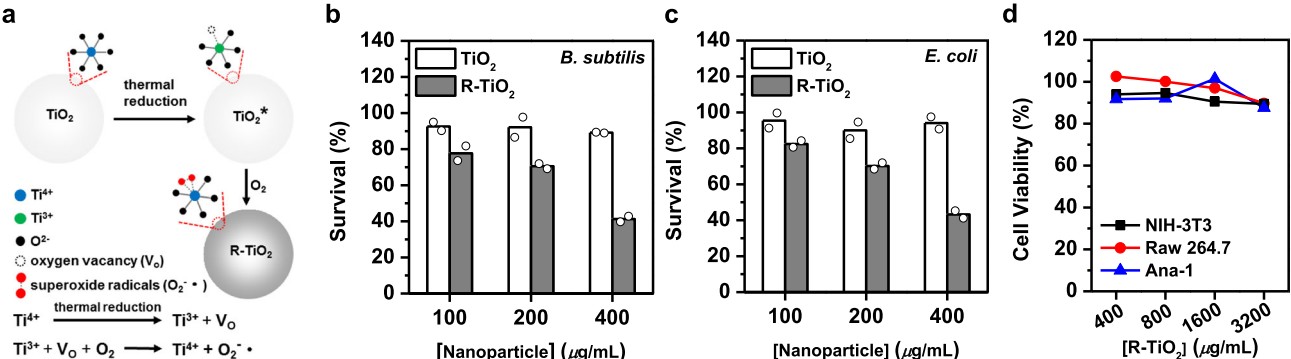

**Fig. 4 Another surface-bound ROS-generating nanozyme with selectivity. a** Schematic illustration on the preparation of thermally reduced $TiO_2$ nanoparticle (i.e., $R\text{-}TiO_2$) from $TiO_2$ nanoparticle (i.e., $TiO_2$) and its generation of surface-bound superoxide radical in the dark. **b**, **c** Survival ratios of **b** *B. subtilis* and **c** *E. coli* cells after treatment with $R\text{-}TiO_2$ and $TiO_2$. **d** Viability ratios of NIH-3T3, Raw 264.7, and Ana-1 cells after $R\text{-}TiO_2$ treatment. Each in vitro antibacterial and cell viability assay was carried out in triplicate and the reported results are averages of two independent trials. Plots on source data are provided as Supplementary Figures. Source data are provided in the Source Data file.

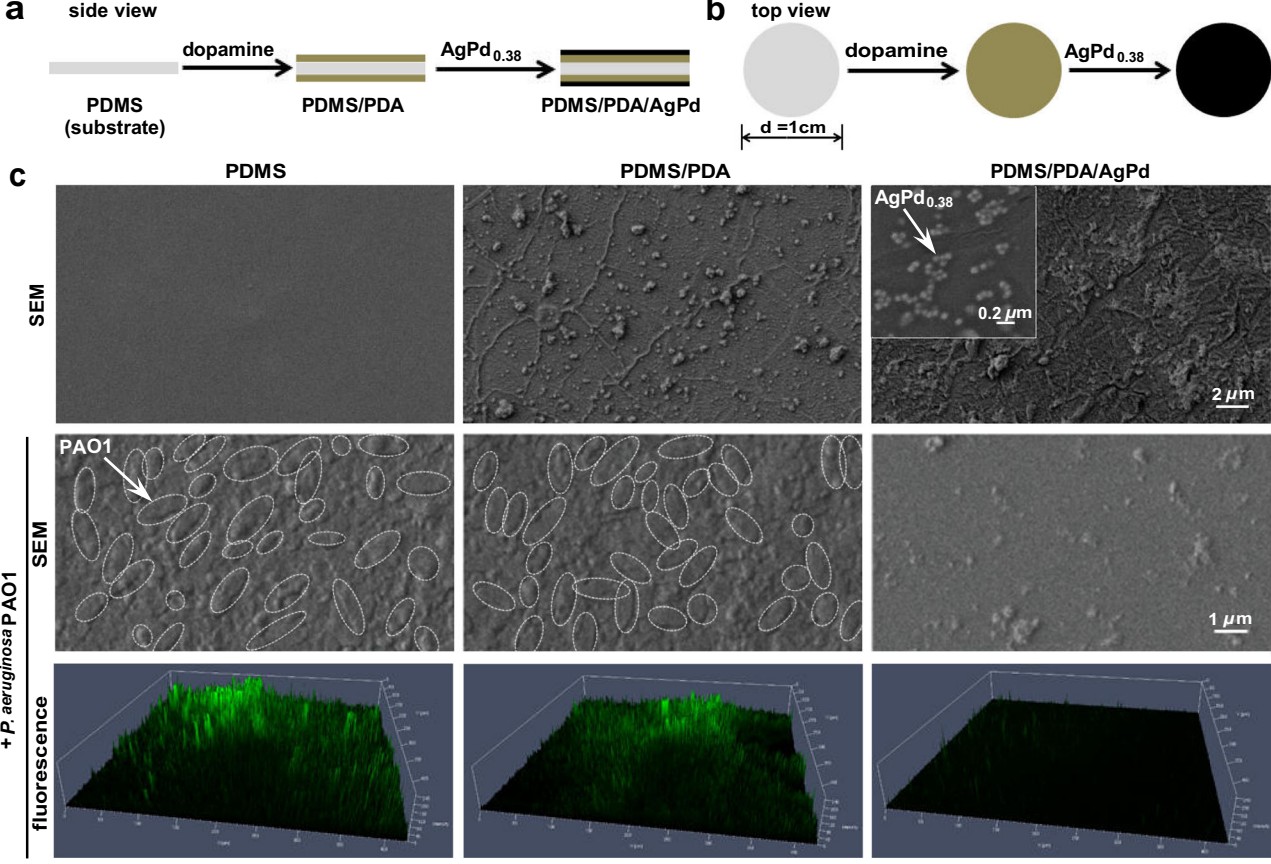

**Fig. 5 AgPd$_{0.38}$ as coating additive enabled an inert surface to inhibit biofilm formation in vitro. a**, **b** Schematic illustration on the preparation of AgPd$_{0.38}$-coated PDMS surface (i.e., PDMS/PDA/AgPd), by coating a circular PDMS wafer successively with dopamine and AgPd$_{0.38}$. **c** (1st row) SEM images of (from left to right) the pristine PDMS surface, the intermediate PDMS/PDA surface, and the final PDMS/PDA/AgPd surface. (Inset) Zoomed-out SEM image of PDMS/PDA/AgPd, with a white arrow indicating an AgPd$_{0.38}$ particle. (2nd row) SEM images of (from left to right) PDMS, PDMS/PDA, and PDMS/PDA/AgPd surfaces after a 5-day incubation with *P. aeruginosa* PAO1 in nutrient broth. Ellipses outlined by white dashed lines indicate intact *P. aeruginosa* PAO1 cells. *P. aeruginosa* PAO1 is a Gram-negative bacterium that is naturally green fluorescent and has a strong tendency to form biofilms. (3rd row) Reconstructed 3-dimensional confocal fluorescence microscopy images of *P. aeruginosa* PAO1 cells on (from left to right) PDMS, PDMS/PDA, and PDMS/PDA/AgPd after 5-day co-incubation in nutrient broth.

*P. aeruginosa* PAO1 cells on PDMS and PDMS/PDA, but not on PDMS/PDA/AgPd. Of note, the brightly green fluorescent substances on PDMS and PDMS/PDA surfaces appeared to be in 3-dimensional irregular structures, indicative of biofilm formation.

Obviously, PDMS/PDA/AgPd, but neither PDMS nor PDMS/PDA, inhibited biofilm formation by *P. aeruginosa* PAO1, owing to the presence of AgPd$_{0.38}$. Taken together, these results suggest that, as a surface-coating additive, AgPd$_{0.38}$ effectively endows a

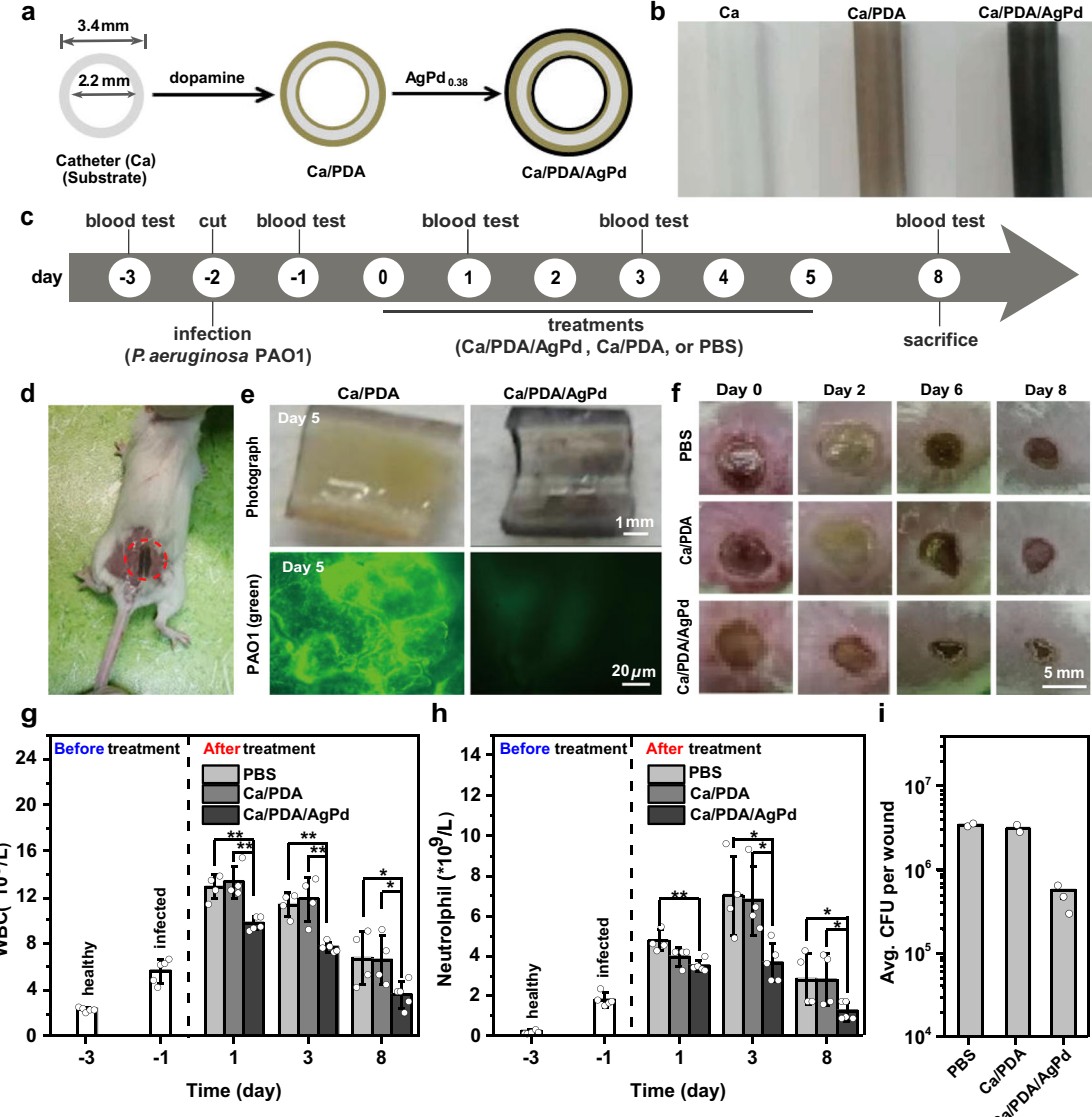

**Fig. 6 AgPd$_{0.38}$ as a coating additive enabled an inert catheter to biocompatibly inhibit biofilm formation in mouse models. a** Schematic illustration on the preparation of Ca/PDA/AgPd, which was done by a coating of a catheter (abbreviated as "Ca") (2.2/3.4 mm in inner/outer diameters) successively with dopamine and AgPd$_{0.38}$. **b** Photographs of (from left to right) the pristine Ca, the intermediate Ca/PDA, and the final Ca/PDA/AgPd catheters. **c** Schedule of the test on the performance of Ca/PDA/AgPd inserted topically in *P. aeruginosa* PAO1-infected wounds in mouse models. Ca/PDA and PBS were included as controls. **d** Photograph of a Ca/PDA/AgPd inserted topically in a *P. aeruginosa* PAO1-infected wound in a mouse model. **e** (Top) Photograph and (bottom) confocal fluorescence microscopy image of a Ca/PDA/AgPd catheter collected carefully after 5-day insertion in a PAO1-infected wound therein and then cut open along its longitudinal axis, with those of Ca/PDA-treated similarly included for comparison. **f** Photographs of wounds treated with (from top to bottom) PBS, Ca/PDA, and Ca/PDA/AgPd throughout the observation window. **g, h** Serum levels of **g** white blood cell (WBC) and **h** neutrophil in mice from different treatment groups on differing days. Data points are reported as mean ± standard deviation ($n = 4$ for PBS or Ca/PDA treatment, $n = 5$ for others). * and ** indicate $p < 0.05$ and $p < 0.01$, respectively, analyzed by two-sided Student's *t*-test. **i** Average CFU per wound for different treatment groups on day 8. Data points are reported as mean ± standard deviation. Source data are provided in the Source Data file.

substrate with the ability to repel protein adsorption, to kill bacteria, and to inhibit biofilm formation.

To examine whether AgPd$_{0.38}$-coated surface effectively inhibits biofilm formation under the complex pathological conditions in vivo, we used a catheter (abbreviated as "Ca") regularly available in clinics as a model for biomedical devices, coated it with AgPd$_{0.38}$ (at $14.61 \pm 1.22$ µg/cm$^2$) (Fig. 6a, b and Supplementary Fig. 55), and examined the performance of the resulting Ca/PDA/AgPd catheter in mouse models bearing wounds infected with *P. aeruginosa* PAO1 (Fig. 6c). For comparison purposes, the nanocage-free intermediate catheter, Ca/PDA, and PBS were examined under comparable conditions; the precursor catheter

was neglected, as catheter-related infections occur regularly in clinics[60] after all. Specifically, the catheters were inserted into the pre-infected wounds (Fig. 6d), left there for 5 days, and then removed from wounds and cut carefully along the longitudinal axis for subsequent examinations. Surprisingly, the as-cut catheters appeared distinct even to naked eyes, with Ca/PDA/AgPd apparently clean while Ca/PDA covered by mucous substances (Fig. 6e, 1st row). When they were subsequently examined under fluorescence microscopy, Ca/PDA/AgPd exhibited a negligible extent of green fluorescence whereas Ca/PDA appeared intensely green fluorescent despite that no staining process was applied at all (Fig. 6e). Clearly, even in the complex pathological

environment in vivo, $AgPd_{0.38}$ as a surface coating additive successfully conferred an otherwise inert substrate with the ability to inhibit biofilm formation, owing to its oxidase-like activity which endows antifouling and antibacterial capacities. Intriguingly, the observed ability to remain biofilm-inhibitive in a complex environment is not exclusive to $AgPd_{0.38}$ but appears to be universal for ROS-generating nanozymes, as $V_2O_5$ nanoparticle, a haloperoxidase mimicry, coated on a substrate thwarts biofilm formation even in ocean[10].

From the perspective of the as-tested mouse models, Ca/PDA/AgPd, but not Ca/PDA, conferred appreciable benefits including promoted wound healing, undetectable off-target toxicity, suppressed immune response, and reduced bacterial burdens (Fig. 6f–i and Supplementary Fig. 56). Throughout the observation window, Ca/PDA/AgPd, but not Ca/PDA, brought about faster shrinkage in wound size than PBS (Fig. 6f), indicative of significantly promoted wound healing owing to the presence of $AgPd_{0.38}$. Meanwhile, Ca/PDA/AgPd-treated mice exhibited stable average body weights as did the Ca/PDA and PBS-treated ones (Supplementary Fig. 56a), indicative of lack of acute toxicity despite the presence of $AgPd_{0.38}$. Host immune responses due to skin infections are commonly reflected as enhanced counts of white blood cell (WBC) and neutrophil in blood[61]. Indeed, significantly higher blood counts of WBC and neutrophil were observed in PAO1-infected mice (i.e., day −1) than in healthy ones (i.e., day −3) (Fig. 6g, h). In the absence of effective treatment, as was the case with PBS, the blood counts of WBC and neutrophil continued to increase in the next days, reaching their peak levels on day 1 and day 3, respectively, and then dropping to what slightly above those in freshly infected mice (i.e., day −1), likely thanks to the innate immunity of mouse. Intriguingly, on any test day, Ca/PDA/AgPd-treated mice exhibited significantly lower counts of WBC and neutrophil than PBS-treated ones (Fig. 6g, h); in stark contrast, in Ca/PDA-treated mice, their relationship of WBC counts versus time was nearly identical to that in PBS-treated mice and same was true with their relationship of neutrophil versus time. Apparently, Ca/PDA/AgPd, but not Ca/PDA, suppressed the inflammatory response caused by a bacterial infection, owing to the presence of $AgPd_{0.38}$. On day 8 (when the observation was stopped), Ca/PDA/AgPd-treated wounds exhibited significantly lower bacterial burdens (gauged in average CFU per wound) than Ca/PDA-treated ones, which instead revealed bacterial burdens comparable to those in PBS-treated mice, indicative of promoted disinfection owing to the presence of $AgPd_{0.38}$. Likely owing to the above benefits, all mice treated with Ca/PDA/AgPd survived on day 8 (when the observation was stopped), and indicative of 100% survival, as compared to the 80% survival of those treated with either Ca/PDA or PBS (Supplementary Fig. 56b). Collectively, these results suggest $AgPd_{0.38}$ as a promising coating additive not only for inhibiting biofilm formation under the complex pathological conditions in vivo but also for benefiting the as-treated host (e.g., by reducing their bacterial burdens and suppressing their immune response).

In summary, we demonstrate the feasibility of using nanozymes that generate surface-bound ROS to kill selectively bacteria over mammalian cells. The result was derived from an analysis of three distinct nanozymes that universally produce surface-bound ROS, with an oxidase-like silver-palladium bimetallic alloy nanocage, $AgPd_{0.38}$, being the lead one. The selectivity is attributable to the surface-bound nature of the as-generated ROS and an unexpected antidote role of endocytosis, a cellular process that is common to mammalian cells but absent in bacteria; in fact, nanoparticles that generate diffusive ROS are toxic indiscriminately to bacteria and mammalian cells, and inhibiting—rather than facilitating—endocytosis sensitizes mammalian cells to $AgPd_{0.38}$. Though generating surface-bound ROS, $AgPd_{0.38}$ efficiently eliminated antibiotic-resistant bacteria and effectively delayed the onset of bacterial resistance emergence. When used as a coating additive, $AgPd_{0.38}$ enabled an originally inert surface to inhibit biofilm formation and suppress the infection-related immune responses in mouse models. This work opens an avenue toward biocompatible nanozymes and may have implications in our fight against both genetically encoded and phenotypic AMR.

## Methods

**Materials**. Trisodium citrate was purchased from J&K Scientific (Beijing, China). Gold nanoparticles (citrate) (AuNPs) were purchased from NanoComposix (USA). Poly(lactide-co-glycolide) (75:25)-b-poly(ethylene glycol) ($PLGA_{40k}$-$PEG_{2k}$) (PLGA, average Mn ~40,000 Da; PEG, average Mn ~2000 Da) were purchased from Xi'an ruixi Biological Technology (Shanxi, China). Chlorin e6 (Ce6) was obtained from Frontier Scientific, Inc. (USA). 1,2-dioleoyl-sn-glycero-3-phosphocholine (DOPC) and 1,2-distearoyl-sn-glycero-3-phosphoethanolamine-N-(folate(polyethylene glycol)-2000) ($DSPE$-$PEG_{2k}$) were purchased from Avanti Polar Lipids (Alabama, USA). Ascorbic acid (AA) was purchased from Sinopharm Chemical Reagent Company (Shanghai, China). Phosphate buffer saline (PBS) (1.35 M NaCl, 47 mM KCl, 100 mM $Na_2HPO_4$, 20 mM $NaH_2PO_4$, pH = 7.4), cell counting kit-8 (CCK-8), ELISA kit, Ethylene Diamine Tetraacetic Acid (EDTA), BCA kit and protease inhibitor were purchased from Beyotime Biotechnology (Shanghai, China). (9,10-anthracenediyl-bis(methylene) dimalonic acid) (ABDA), Ethylene glycol, prostaglandin E1, and poly(vinylpyrrolidone) (PVP) were purchased from Sigma-Aldrich (Shanghai, China). SYLGARD 184 elastomer kit was purchased from Dow Corning (Shanghai, China). $TiO_2$ (Degussa, P25) was purchased from Lijie chemicals (Shaoxing, China). Singlet oxygen sensor green (SOSG) was purchased from Thermo Fisher (Shanghai, China). PTA (p-phthalic acid) was purchased from ShangHai YuanYe Biotechnology (Shanghai, China). Mueller–Hinton (MH) broth powder and tryptic soy broth (TSB) were purchased from Qingdao Hope BioTechnology (Qingdao, China). Live/Dead BacLight bacterial viability kit was purchased from Molecular Probes (Shanghai, China). Dynasore, gentamicin, levofloxacin, sodium hydrosulfide (NaHS), silver trifluoroacetate ($CF_3COOAg$), potassium chloropalladite ($K_2PdCl_4$), and verteporfin (Ver) were purchased from Aladdin-Reagent (Shanghai, China). Fetal bovine serum (FBS) was purchased from Shanghai ExCell Biology. Inc. Dulbecco's modified Eagle's medium (DMEM) was purchased from Hyclone. Bacterial strains used in this work were purchased from the American Type Culture Collection (ATCC) (Virginia, USA). All cells used in this work were purchased from the Cell Bank of the Chinese Academy of Sciences (Shanghai, China). All other reagents were purchased from Sinopharm Chemical Reagent Company (Shanghai, China). All reagents were used as received without further purification unless specified otherwise.

**TEM characterizations on the mixtures of AuNPs and _E. coli_ cells**. TEM samples were prepared by a typical process of fixation, dehydration, and embedding in a resin matrix according to the reported method[62]. _E. coli_ (ATCC 25922) was used as a representative strain. Briefly, 3–5 individual colonies were inoculated into fresh soy broth (TSB) and incubated at 37 °C for 16–18 h to stationary phase. A 40 μL culture was diluted with fresh TSB by 100-fold and regrown at 37 °C to mid-log phase ($OD_{600}$ = 0.5–0.7). Bacterial cells were then harvested and washed once with sterile PBS via centrifugation (10,000 × g for 5 min at 4 °C), adjusted with sterile PBS to ~3 × $10^8$ CFU/mL. An AuNP (5, 10, or 20 nm) dispersion in sterile PBS buffer (0.5 mL) was added into 0.5 mL adjusted bacteria suspension, to achieve a final bacterial inoculum size of ~1.5 × $10^8$ CFU/mL and a final nanoparticle mass concentration of 50 μg/mL. The resulting mixture was then incubated at 37 °C for 2 or 6 h and subsequently subjected to centrifuge at 10,000 × g for 5 min to remove the supernatant, followed by fixation with 5% glutaraldehyde for 2 h at room temperature, then successively fixed at 4 °C overnight. Fixed bacterial cells were washed thrice with sterile PBS (0.1 M, pH = 7.2), and stained with a solution of 2% osmium tetroxide (1 mL) overnight. In the following, the sample was rinsed thrice with sterile PBS (0.1 M, pH = 7.2) (20 min each, 1 mL), and then dehydrated in a series of graded ethanol solutions (30, 50, 70, 80, 95, and 100% ethanol in water), then rinsed thrice with acetone (20 min each, 1 mL). The resulting sample was incubated in resin solution for 1 h (with resin-to-acetone volume ratio at 1:1) and then for 3 h (with resin-to-acetone volume ratio at 3:1). The sample was then embedded in a fresh 100% resin for ~36 h. A new batch of resin (250 μL) was added, and the sample was cured at 70 °C for two days. Finally, 70-nm-thick samples were sliced off using an Ultramicrotome equipped with a diamond knife (Leica UC7) and placed on copper grids for TEM imaging.

**Preparation of Ag nanoparticles, Ag nanocubes, and AgPd nanocages**. Ag nanoparticles were synthesized by means of using citrate as a surfactant and ascorbic acid as reductant. Typically, in a 100 mL flask, a 40.0 mL aqueous solution containing trisodium citrate (3.0 × $10^{-3}$ M) and ascorbic acid (6.0 × $10^{-4}$ M) was adjusted to pH value 11 by slowly adding 0.1 mol/L NaOH solution. The 0.4 mL $AgNO_3$ aqueous solution (0.1 M) was added into the flask under a stirring speed of 1000 rpm in a 35 °C water bath. The color of the reaction solution was immediately changed from colorless to deep brown. After 15 min, the reaction was terminated

by flowing cold water for several minutes. Finally, the Ag nanoparticles were centrifuged (10 min at 850 × $g$) and washed with deionized water several times to remove excess surfactant and impurities.

Ag nanocubes were synthesized by following procedures with slight modifications based on previous protocol[27]. In a typical synthesis, 20 mL ethylene glycol (EG) was added into a 50-mL-round-bottom flask and heated to 150 °C under magnetic stirring in an oil bath for one hour. Sodium hydrosulfide (NaHS) (3 mM, 0.24 mL) was firstly injected into the heated solution, and after 4 min, HCl solution (3.5 mM, 2 mL) was added, followed by the addition of poly (vinylpyrrolidone) (PVP, 20 mg/mL, 5 mL, MW = 55,000). Another 2 min later, silver trifluoroacetate (CF$_3$COOAg, 282 mM, 1.6 mL) was quickly added into the mixture. All reagents were freshly made in ethylene glycol. The edge length of Ag nanocubes could be well controlled by changing the reaction time and monitoring the LSPR peak position measured by UV-vis spectrometer. After a period of time, the solution was quenched by soaking it into an ice-water bath without stopping stirring. Finally, the resulting samples were collected and purified with ethanol and DI water several times by centrifugation, then re-dispersed in DI water for further use.

AgPd nanocages were synthesized by the process of the galvanic replacement reaction between Ag nanocubes and Potassium chloropalladite (K$_2$PdCl$_4$). DI water (20 mL) containing 200 mg PVP was heated to 90 °C in a 50 mL round-bottom flask connected with a reflux condenser under magnetic stirring. The aqueous solution of Ag nanocubes (1 mg/mL, 0.5 mL) was added into the flask and 10 min later, K$_2$PdCl$_4$ (0.5 mM) was slowly dropped into the solution at the rate of 0.4 mL/min by using an injection pump. The different molar ratios of the AgPd hollow nanostructures are related to the volume and concentration of K$_2$PdCl$_4$, as well as the dropping rate. The reaction was terminated by placing the flask in an ice-water bath and kept stirring during the whole process. A small amount of solid KCl was sprinkled into the solution until the AgCl was dissolved and removed with the supersaturation of Cl$^-$, which could help to yield a well-defined hollow structure of nanocages. The resulting solution was washed five times with DI water by centrifugation, and then re-dispersed in DI water for further use.

To examine a nanoparticle's morphology, a drop of the aqueous suspension of the nanoparticle was added onto a piece of the carbon-coated copper grid, dried under ambient conditions, and then viewed under a transmission electron microscope (TEM) (Hitachi H-7700 operated at 100 kV, and JEOL JEM-2100F operated at 200 kV for EDS mapping and HRTEM).

To measure the elemental composition of a nanoparticle, the nanoparticle was dissolved with aqua regia (HCl/HNO$_3$ = 3:1, volume ratio) and the resulting solution was then subjected to quantification on metal contents with an inductively coupled plasma mass spectrometry (ICP-MS) (PlasmaQuad 3, Thermo Scientific).

To confirm the structure of a nanoparticle, a nanoparticle was further characterized by X-ray powder diffraction (XRD). XRD patterns were recorded by using a Philips X'Pert Pro Super X-ray diffractometer with Cu-K$\alpha$ radiation ($\lambda$ = 1.5418 Å).

To monitor the size and surface zeta potential of a nanoparticle, the nanoparticle was dispersed into Millipore water to a final concentration of 10 μg/mL, and the resulting dispersion was subsequently subjected to dynamic light scattering measurement with a nanoparticle analyzer (Nano ZS90, Malvern). A similar protocol was followed to monitor the nanoparticle's colloidal stability in 1× PBS or Mueller–Hinton (MH) broth over a span of 72 h.

**Preparation of thermally reduced TiO$_2$ nanoparticle (R-TiO$_2$).** The R-TiO$_2$ was prepared by following a previously reported protocol[54,55] with slight modifications. Briefly, polycrystalline TiO$_2$ nanoparticle (Degussa P25) powder (10 mg) was slowly heated (~5 h) under vacuum to a maximum temperature of 500 °C and held at 500 °C for 1 h. The as-reduced powder was then cooled naturally to room temperature, followed by exposure to an excess amount of air, which yielded the expected R-TiO$_2$.

The morphology and size of the resulting R-TiO$_2$ nanoparticle were examined with TEM, with its precursor TiO$_2$ nanoparticle (Degussa P25) included for reference. Briefly, a nanoparticle (TiO$_2$ or R-TiO$_2$) was dispersed into Millipore water to a final concentration of 100 μg/mL, and the resultant dispersion (2 drops) was added onto a copper grid, dried in an oven at 37 °C for 30 min, and then imaged under a TEM (Hitachi H-7650, operated at 100 kV).

To examine whether the as-generated ROS species include superoxide radicals (O$_2^-$•), we carried out fluorescence-based ROS detection assays but using Dihydroethidium (DHE). Specifically, DHE solution in PBS (4 μL, 5 mM) was mixed with a nanoparticle dispersion in PBS (at an expected mass concentration and volume of 200 μL), followed by incubation at 37 °C for 3 h. The resulting mixture was subsequently centrifuged at 8000 × $g$ for 5 min (Eppendorf, Centrifuge 5417R) to remove the nanoparticle, and the resulting supernatant was then subjected to fluorescence emission spectrum recording ($\lambda_{ex}/\lambda_{em}$ = 480 nm/500–700 nm, slit-widths of 10 nm for both excitation and emission wavelengths) with a fluorimeter (F-4600 spectrofluorometer, Hitachi). Control is DHE in PBS-treated similarly but without any nanoparticle. The reported results are averages of two independent trials.

**Preparation of Ce6/PLGA or Ver/PLGA nanoparticle.** PLGA$_{40k}$-PEG$_{2k}$ nanoparticle preloaded with Chlorin e6 (Ce6) (i.e.,Ce6/PLGA) or verteporfin (Ver) (i.e.,

Ver/PLGA) was prepared by a nanoprecipitation process[21]. Briefly, into a PLGA$_{40k}$-PEG$_{2k}$ solution (1 mL, 3 mg/mL in acetone) was added the solution of Ce6 or Ver (1 mL, 0.15 mg/mL in acetone). The resulting mixture was added dropwise into sterile Millipore water (3 mL), followed by stirring (at 300 rpm) in the open air for 24 h to evaporate the acetone, which yielded the dispersion of the as-expected Ce6/PLGA or Ver/PLGA nanoparticle.

The morphology and size of the resulting nanoparticles were examined with TEM. Briefly, Ce6/PLGA or Ver/PLGA nanoparticle was dispersed into Millipore water to a final concentration of 100 μg/mL, and the resultant dispersion (2 drops) was added onto a copper grid, negatively stained with a solution of phosphotungstic acid (1 wt% in PBS) (1–2 drops), dried in an oven (at 37 °C), and then imaged under a TEM (Hitachi H-7650 operated at 100 kV).

The hydrodynamic diameter and the zeta potential (ζ-potential) of as-expected nanoparticles were measured by performing dynamic light scattering (DLS) characterizations. Briefly, a particle was dispersed into Millipore water to a final concentration of 100 μg/mL, and the resulting dispersion was subjected to DLS characterizations using particle size and zeta potential analyzer (Zetasizer Nano ZS90, Malvern).

**Preparation of PEGylated liposome.** DOPC stock solution was mixed with DSPE-PEG$_{2k}$ stock solutions at a mass ratio of DOPC: DSPE-PEG$_{2k}$ = 90:10; all lipid stock solutions were at 20 mg/g in chloroform. The resulting mixtures were dried to a thin film under a gentle stream of N$_2$, and desiccated under vacuum overnight. The resulting thin film was rehydrated with Millipore water at 40 °C for 2 h to a final lipid concentration of 10 mg/mL. The resulting lipid dispersion was subjected to seven freeze–thaw cycles and subsequently extruded through a nucleopore membrane with a pore size of 0.2 μm (Whatman) for 13 times using mini-extruder (Avanti Polar Lipids), which yielded the expected PEGylated liposome. All lipid dispersions were stored at 4 °C prior to use.

The hydrodynamic diameter and ζ-potential of the as-prepared liposome were measured by performing DLS characterizations. Briefly, liposome dispersion (100 μg/mL) was subjected to DLS characterizations using particle size and zeta potential analyzer (Zetasizer Nano ZS90, Malvern).

**Preparation of platelet membrane vesicles.** Platelet membrane vesicles were derived from platelets through a protocol we previously reported[21]. Briefly, fresh whole blood was collected from female mice (ICR) (6–8 wk old) (Beijing Vital River Laboratory Animal Technology Co., Ltd) into a plain tube specifically designed for blood collection (Rich Science Industry Co., Ltd, Chengdu, China), followed by centrifugation (Eppendorf, 5417R) at 200 × $g$ for 20 min, which separated the platelet-rich plasma and the white and red blood cells respectively into the supernatant, the middle layer, and the bottom layer. The resulting supernatant was subsequently transferred into a sterile 5-mL centrifuge cup, followed by the addition of PBS buffer supplemented with 1 mM of EDTA and 2 μM of prostaglandin E1—which are to prevent platelet activation—and then centrifugation (Eppendorf, Centrifuge 5417R) at 800 × $g$ for 20 min at room temperature to collect platelets. The resulting pellet (i.e., platelets) was collected and then re-suspended into PBS (2 mL) supplemented with 1 mM of EDTA and protease inhibitor (Beyotime, China). The resulting platelet suspension was subsequently subjected to three freeze–thaw cycles (frozen at −80 °C and then thawed at room temperature) and then centrifugation at 4000 × $g$ for 3 min at room temperature and re-dispersion into PBS supplemented with protease inhibitor, which yielded the platelet ghosts. The resulting platelet ghost dispersion was subsequently washed for three times with PBS supplemented with protease inhibitor, re-dispersed into sterile Millipore water, and sonicated for 5 min using a water bath sonicator (KUDOS, SK5210HP) at a frequency of 53 kHz and output power of 100 W, which yielded platelet membrane vesicles dispersed in Millipore water. The as-prepared dispersion of platelet membrane vesicles was stored at −80 °C prior to use.

**Preparation of Ce6/PLGA @lipid or Ver/PLGA@lipid nanoparticle.** Coating the Ce6/PLGA or Ver/PLGA nanoparticle with a lipid bilayer was accomplished by sonicating the mixture of Ce6/PLGA or Ver/PLGA and liposome (DOPC: DSPE-PEG$_{2k}$ = 90:10) (mass ratio of lipid to PLGA = 1:1) in a bath sonicator (KUDOS, SK5210HP) at a frequency of 53 kHz and output power of 100 W for 5 min, followed by centrifugation (5417R, Eppendorf) at 10,000 × $g$ for 10 min to remove excess liposome, which yielded the as-expected lipid bilayer-coated nanoparticle (namely, Ce6/PLGA@lipid or Ver/PLGA@lipid).

The morphology and size of the resulting nanoparticles were examined with TEM. Briefly, Ce6/PLGA@lipid or Ver/PLGA@lipid nanoparticle was dispersed into Millipore water to a final concentration of 100 μg/mL, and the resultant dispersion (2 drops) was added onto a copper grid, negatively stained with a solution of phosphotungstic acid (1 wt% in PBS) (1–2 drops), dried in an oven (at 37 °C), and then imaged under a TEM (Hitachi H-7650 operated at 100 kV).

The hydrodynamic diameter and ζ-potential of the resulting Ce6/PLGA@lipid or Ver/PLGA@lipid were measured by performing DLS characterizations. Briefly, Ce6/PLGA@lipid or Ver/PLGA@lipid dispersion (100 μg/mL) was subjected to DLS characterizations using particle size and zeta potential analyzer (Zetasizer Nano ZS90, Malvern).

**Preparation of AgPd$_{0.38}$@lipid nanoparticle**. Coating AgPd$_{0.38}$ with a lipid bilayer was accomplished by sonicating the mixture of AgPd$_{0.38}$ and liposome (DOPC: DSPE-PEG$_{2k}$ = 90:10) (mass ratio of lipid to AgPd$_{0.38}$ = 1:1) in a bath sonicator (KUDOS, SK5210HP) at a frequency of 53 kHz and output power of 100 W for 5 min, followed by centrifugation (5417R, Eppendorf) at 8000 × g for 10 min to remove excess liposome, which yielded the as-expected lipid bilayer-coated AgPd$_{0.38}$ (namely, 5417).

The morphology and size of the resulting AgPd$_{0.38}$@lipid were examined with TEM. Briefly, AgPd$_{0.38}$@lipid was dispersed into Millipore water to a final concentration of 100 μg/mL, and the resultant dispersion (1 drop) was added onto a copper grid, negatively stained with a solution of phosphotungstic acid (1 wt% in PBS) (1–2 drops), dried in an oven (at 37 °C), and then imaged under a TEM (Hitachi H-7700 operated at 100 kV).

The hydrodynamic diameter and ζ-potential of AgPd$_{0.38}$@lipid were measured by performing dynamic light scattering (DLS) characterizations. Briefly, AgPd$_{0.38}$@lipid was dispersed into Millipore water to a final concentration of 10 μg/mL, and the resulting dispersion was subjected to DLS characterizations using particle size and zeta potential analyzer (Zetasizer Nano ZS90, Malvern).

**Preparation of AgPd$_{0.38}$@P nanoparticle**. Coating AgPd$_{0.38}$ with platelet membrane was accomplished by sonicating the mixture of AgPd$_{0.38}$ and platelet membrane vesicle (mass ratio of lipid to AgPd$_{0.38}$ = 1:1) in a bath sonicator (KUDOS, SK5210HP) at a frequency of 53 kHz and output power of 100 W for 5 min, followed by centrifugation (5417R, Eppendorf) at 8000 × g for 10 min to remove excess platelet membrane, which yielded the as-expected platelet membrane-coated AgPd$_{0.38}$ (namely, AgPd$_{0.38}$@P).

The morphology and size of the resulting AgPd$_{0.38}$@P were examined with TEM. Briefly, AgPd$_{0.38}$@P was dispersed into Millipore water to a final concentration of 100 μg/mL, and the resultant dispersion (1 drop) was added onto a copper grid, negatively stained with a solution of phosphotungstic acid (1 wt% in PBS) (1–2 drops), dried in an oven (at 37 °C), and then imaged under a TEM (Hitachi H-7700 operated at 100 kV).

The hydrodynamic diameter and ζ-potential of AgPd$_{0.38}$@lipid were measured by performing dynamic light scattering (DLS) characterizations. Briefly, AgPd$_{0.38}$@P was dispersed into Millipore water to a final concentration of 10 μg/mL, and the resulting dispersion was subjected to DLS characterizations using particle size and zeta potential analyzer (Zetasizer Nano ZS90, Malvern).

**Monitoring ROS generation with ascorbic acid (AA) as the probe**. To examine whether a nanoparticle is able to mimic oxidases, we measure the production of reactive oxygen species (ROS) in an aqueous dispersion of the nanoparticle, using ascorbic acid (AA) as the ROS probe. AA shows absorption at 266 nm but can be oxidized by ROS to form dehydroascorbic acid[13]. Into phosphate buffer solution (PBS, at pH 7.4) were successively added dispersion of a nanoparticle dispersion in PBS (to an expected final particle mass concentration of 8, 16, or 32 μg/mL) and AA solution in PBS (to a final AA concentration of 60 μM); The final volumes of the resulting dispersions were kept constant at 1 mL. The resulting dispersions were subsequently incubated at 37 °C for 1 h or 3 h, followed by centrifuge (Eppendorf, Centrifuge 5417R) at 8000 × g for 10 min to remove the nanoparticles. The resultant supernatant was then subjected to ultraviolet-visible absorption spectrum recording with a UV-vis spectrometer (Cary 60 UV-Vis, Agilent). Control is an AA solution (60 μM, 1 mL).

**Monitoring the generation of singlet oxygen ($^1O_2$)**. To examine whether the as-generated ROS species include singlet oxygen, we carried out similar fluorescence-based ROS detection assays but using 9,10-anthracenediyl-bis(methylene) dimalonic acid (ABDA) as the probe. ABDA is a fluorescent probe but, upon oxidization selectively by singlet oxygen, becomes non-fluorescent[21]. Briefly, ABDA solution in PBS (100 μL, 40 μM) was mixed with the dispersion of a nanocage in PBS (100 μL, at varying mass concentration of nanoparticle), followed by incubation at 37 °C for 3 h. The resulting mixture was subsequently centrifuged at 8000 × g for 10 min (Eppendorf, Centrifuge 5417R) to remove the nanoparticles, and the resulting supernatant was then subjected to fluorescence emission spectrum recording ($\lambda_{ex}$/$\lambda_{em}$ = 380 nm/400–600 nm, slit-widths of 2.5 nm for both excitation and emission wavelengths) with a fluorimeter (F-4600 spectrofluorometer, Hitachi). Control is ABDA in PBS-treated similarly but without any nanoparticle. The generation of singlet oxygen was indicated by the relative decrease in the fluorescence intensity at 433 nm of a nanoparticle-containing dispersion compared to that of ABDA in PBS, which is relative fluorescence intensity (%) = ($I_{(nanoparticle+ABDA)}$/$I_{ABDA}$) × 100%. The reported results are averages of two independent trials.

To confirm the generation of singlet oxygen, we carried out similar assays but replacing ABDA with SOSG (singlet oxygen sensor green) (Thermo Fisher). SOSG emits weak blue fluorescence ($\lambda_{ex}$ = 372, 393 nm; $\lambda_{em}$ = 395, 416 nm) but upon oxidization selectively by singlet oxygen becomes the endoperoxide of SOSG (SOSG-EP), which emits bright green fluorescence ($\lambda_{ex}$/$\lambda_{em}$ = 504 nm/525 nm)[31]. Specifically, SOSG solution in PBS (20 μL, 50 μM) was mixed with a nanoparticle dispersion in PBS (180 μL, at varying mass concentrations of nanoparticles), followed by incubation at 37 °C for 3 h. The resulting mixture was subsequently centrifuged at 8000 × g for 10 min (Eppendorf, Centrifuge 5417R) to remove the

nanoparticles, and the resulting supernatant was then subjected to fluorescence emission spectrum recording ($\lambda_{ex}$/$\lambda_{em}$ = 504 nm/510–700 nm, slit-widths of 5 nm for both excitation and emission wavelengths) with a fluorimeter (F-4600 spectrofluorometer, Hitachi). Control is SOSG in PBS-treated similarly but without any nanoparticle. The generation of singlet oxygen was indicated by the relative enhancement in fluorescence at 525 nm of a nanoparticle-containing dispersion compared to that of SOSG in PBS, which is relative fluorescence intensity (%) = ($I_{(nanoparticle+SOSG)}$/$I_{SOSG}$) × 100%. The reported results are averages of two independent trials.

**Monitoring the generation of hydroxyl radical**. To assess whether the as-generated ROS species include hydroxyl radical, we performed similar fluorescence-based ROS detection assays but using p-phthalic acid (PTA) as the probe for hydroxyl radical[33]. PTA is virtually non-fluorescent but, upon oxidation selectively by hydroxyl radical, becomes brightly green fluorescent ($\lambda_{ex}$/$\lambda_{em}$ = 315 nm/320–600 nm). Specifically, PTA powder was dissolved in NaOH aqueous solution (0.2 M), to a PTA concentration of 50 mM. The resulting TPA solution was then mixed with a nanoparticle dispersion in PBS (at an expected mass concentration of nanoparticle) to a final mixture volume of 200 μL and a final TPA concentration of 500 μM, followed by incubation at 37 °C for 3 h. The resulting mixture was subsequently centrifuged at 8000 × g for 10 min to remove the nanoparticles, and the resulting supernatant was then subjected to fluorescence emission spectrum recording ($\lambda_{ex}$/$\lambda_{em}$ = 315 nm/320–600 nm, with slit-widths of 5 and 10 nm for excitation and emission wavelengths, respectively) using a fluorimeter (F-4600 spectrofluorometer, Hitachi). Control is PTA in PBS-treated similarly but without any nanoparticle. Hydroxyl radical generation was indicated by the relative enhanced fluorescence intensity at 400 nm of a nanoparticle-containing dispersion compared to that of PTA in PBS. The reported results are averages of two independent trials.

**Effects of oxygen removal on ROS generation**. To determine how oxygen removal affects a nanoparticle dispersion's ROS production, we performed similar ROS detection assays as described above but, prior to mixing AA solution with the nanoparticle dispersion, purged both the AA solution and the nanoparticle dispersion with Ar or N$_2$ flow for 30 min. The reported results are averages of two independent trials.

**Effects of oxygen pouring on ROS generation**. To determine how oxygen pouring affects a nanoparticle dispersion's ROS production, we performed similar ROS detection assays as described above but, prior to mixing AA solution with the nanoparticle dispersion, poured both the AA solution and the nanoparticle dispersion with O$_2$ flow for 30 min. The reported results are averages of two independent trials.

**Pd and Ag release from AgPd$_{0.38}$ nanocage**. AgPd$_{0.38}$ was dispersed into PBS (pH = 7.4, 1 mL) to a final nanoparticle concentration of 40 μg/mL, followed by incubation at 37 °C for 3 or 18 h, and then centrifuge at 8000 × g for 10 min to remove the nanoparticle. The resulting supernatant (1 mL) was subjected to measurement with inductively coupled plasma mass spectrometry (ICP-MS), to determine the contents of Pd and Ag in the supernatant. The reported results are averages of two independent trials.

**Bacterial plate-killing assays with nanoparticles**. A nanoparticle's bactericidal activity profile was evaluated by performing plate bacterial killing assays. *S. aureus* (ATCC 25923) and *B. subtilis* (ATCC 6051) are used as representative Gram-positive bacterial strains, while *E. coli* (ATCC 25922) and *P. aeruginosa* (ATCC 27853) are used as representative Gram-negative bacterial strains. Moreover, Methicillin-resistant *S. aureus* (MRSA) (ATCC BAA-1720) was used as a representative for antibiotic-resistant bacterial strain. For each bacterial strain, 3–5 individual colonies were inoculated into fresh tryptic soy broth (TSB) and incubated at 37 °C for 16–18 h to stationary phase. A 40 μL culture was diluted with fresh TSB by 100-fold and regrown at 37 °C to mid-log phase (OD$_{600}$ = 0.5–0.7). Bacterial cells were then harvested and washed once with sterile PBS via centrifugation (10,000 × g for 5 min at 4 °C), adjusted with sterile PBS to ~1.5 × 10$^6$ CFU/mL, and inoculated into zero-dilution wells of a preset 96-well microplate. Serial 2-fold dilutions of a nanoparticle dispersion were made with sterile PBS buffer. Each nanoparticle dilution (100 μL) was added into each zero-dilution well in a 96-well microplate. 50 μL adjusted bacteria suspension was inoculated into each zero-dilution well of a preset microplate, to achieve 5 × 10$^5$ CFU/mL in each well (150 μL). The microplate was then incubated at 37 °C for 3 h. Serial 10-fold dilutions were subsequently made with sterile PBS buffer, followed by plating the dilutions (20 μL) onto TSB agar plates for overnight incubation at 37 °C to form visible colonies. Inoculum size was indicated by control samples containing bacteria treated similarly but without nanoparticle. Each trial was performed in triplicate, and the reported results are the averages of two independent trials. The reported minimum bactericidal concentration (MBC) values are defined as the minimum concentrations of antibiotics or nanoparticles to inhibit 99.9% bacterial growth.

**Bacterial plate-killing assays with AgPd$_{0.38}$ in the presence of carotene**. *S. aureus* (ATCC 25923) and *E. coli* (ATCC 25922) are used as representative Gram-positive and Gram-negative bacterial strains. For each bacterial strain, 3–5 individual colonies were inoculated into fresh tryptic soy broth (TSB) and incubated at 37 °C for 16–18 h to stationary phase. A 40 μL culture was diluted with fresh TSB by 100-fold and regrown at 37 °C to mid-log phase (OD$_{600}$ = 0.5–0.7). Bacterial cells were then harvested and washed once with sterile PBS via centrifugation (10,000 × g for 5 min at 4 °C), adjusted with sterile PBS to ~1.5 × 10$^6$ CFU/mL, and inoculated into zero-dilution wells of a preset 96-well microplate. Serial 2-fold dilutions of AgPd$_{0.38}$ dispersion were made with sterile PBS buffer containing carotene (2.25 mg/mL). Each AgPd$_{0.38}$ dilution (100 μL) was added into each zero-dilution well in a 96-well microplate. 50 μL adjusted bacteria suspension was inoculated into each zero-dilution well of a preset microplate, to achieve 5 × 10$^5$ CFU/mL in each well (150 μL). The microplate was then incubated at 37 °C for 3 h. Serial 10-fold dilutions were subsequently made with sterile PBS buffer, followed by plating the dilutions (20 μL) onto TSB agar plates for overnight incubation at 37 °C to form visible colonies. Inoculum size was indicated by control samples containing bacteria treated similarly but without AgPd$_{0.38}$. Each trial was performed in triplicate, and the reported results are the averages of two independent trials.

**SEM characterizations on the bacterial morphology after nanocage treatment**. *E. coli* (ATCC 25922) and *P. aureginosa* (ATCC 27853) were used as representative strains for Gram-negative bacteria, while *S. aureus* (ATCC 25923) and *B. subtilis* (ATCC 6051) stains were used as representative Gram-positive bacteria. Briefly, 3–5 individual colonies were inoculated into fresh tryptic soy broth (TSB) and incubated at 37 °C for 16–18 h to stationary phase. A 40 μL culture was diluted with fresh TSB by 100-fold and regrown at 37 °C to mid-log phase (OD$_{600}$ = 0.5–0.7). Bacterial cells were then harvested and washed once with sterile PBS via centrifugation (10,000×g for 5 min at 4 °C), adjusted with sterile PBS to ~1.5 × 10$^8$ CFU/mL. A nanoparticle dispersion in sterile PBS buffer (84 μL) was added into 166 μL adjusted bacteria suspension, to achieve a final bacterial inoculum size of ~1 × 10$^8$ CFU/mL and a final nanoparticle mass concentration of 20 μg/mL. The resulting mixture was then incubated at 37 °C for 3 h and subsequently subjected to centrifuge at 10,000 × g for 5 min to remove the supernatant, followed by fixation with 4% formaldehyde at 4 °C for 1 h, and then dehydration successively with a series of graded ethanol solutions (25, 50, 75, 90, and 100%) via centrifuge (10,000 × g for 5 min). The resulting pellet was then dispersed in 100 μL ethanol (100%), and part of the resulting dispersion (~10 μL) was dropped on a copper tape and dried overnight. The resulting sample was then sputtered with gold for 20 s and imaged under a scanning electron microscope (SEM) (FEI Apreo, USA). Controls are those assayed similarly but without nanoparticles. The SEM measurements were repeated at least once to confirm that the observations are reproducible.

**Synchrotron soft X-ray microscopy (SXM) assays**. *E. coli* (ATCC 25922) and *S. aureus* (ATCC 25923) were used as the representatives for Gram-negative and -positive bacteria, respectively. For each bacterial strain, 3–5 individual colonies were inoculated into fresh tryptic soy broth (TSB) and incubated at 37 °C for 16–18 h to stationary phase. Bacterial cells were subsequently harvested and washed once with sterile PBS via centrifugation (5417R, Eppendorf) at 10,000 × g for 5 min and diluted with sterile deionized water to a concentration of ~4.5 × 10$^8$ CFU/mL, which yielded the final bacterial inoculum for preparing synchrotron SXM samples. Into a 1.5-mL centrifuge cup were successively added AgNC dispersion (1.5 mg/mL AgNC in 400 μL H$_2$O) and the as-adjusted bacterial inoculum (200 μL). The resulting mixture was subsequently incubated at 37 °C for 10 min and then added dropwise onto a copper grid, immediately followed by freezing in liquid nitrogen and imaging under synchrotron Soft X-ray microscopy (SXM). Controls are bacterial cells assayed similarly but without AgNC.

**Bacterial Dead/Live viability assays with nanoparticles**. Bacterial live/dead viability assays were performed using a bacterial dead/live viability kit, and the staining effects were examined under fluorescence microscopy (IX81, Olympus). SYTO-9 and propidium iodide (PI), two nucleic acid stains with strikingly different spectral characteristics and abilities to permeate healthy bacterial membranes, are used to label all and dead cells, respectively[63]. *E. coli* (ATCC 25922) and *P. aureginosa* (ATCC 27853) were used as representative strains for Gram-negative bacteria, while *S. aureus* (ATCC 25923) and *B. subtilis* (ATCC 6051) stains were used as representative Gram-positive bacteria. For each bacterial strain, 3–5 individual colonies were inoculated into fresh tryptic soy broth (TSB) and incubated at 37 °C for 16–18 h to stationary phase. A 40 μL culture was diluted with fresh TSB by 100-fold and regrown at 37 °C to mid-log phase (OD$_{600}$ = 0.5–0.7). Bacterial cells were then harvested and washed once with sterile PBS via centrifugation (10,000 × g for 5 min at 4 °C), adjusted with sterile PBS to ~1.5 × 10$^8$ CFU/mL. A nanoparticle dispersion in sterile PBS buffer (84 μL) was added into 166 μL the as-adjusted bacteria suspension, to achieve a final bacterial inoculum size of ~1 × 10$^8$ CFU/mL and a final nanoparticle mass concentration of 20 μg/mL. The resulting mixing suspension was then incubated at 37 °C for 3 h. The as-treated bacteria cells (100 μL) were subsequently stained with SYTO-9 (192 μM in PBS, 5 μL) and propidium iodide (PI, 250 μM in PBS, 5 μL) via incubation in the dark for 15 min, and then centrifuged at 10,000 × g for 5 min to remove the supernatant. The bacterial pellets

were then washed with 100 μL PBS. 10 μL of the resultant bacterial suspension was transferred onto a coverslip, air-dried, immersed in PBS (10 μL), and imaged under fluorescence microscopy (IX81, Olympus) using a ×100 oil-immersion objective lens. FITC and TRITC filters were used for SYTO-9 and propidium iodide, respectively. Each sample was imaged at 15 different areas, and quantitative analysis on the microscopy images was performed with ImageJ Software (ImageJ 1.47). Controls are those assayed similarly but without nanoparticles. The reported results have been checked for consistency with two individual samples.

**Outer membrane permeability assays with nanoparticles**. 1-N-Phenylnaphthylamine (NPN) was used to assess the outer membrane permeability due to its weak fluorescent in aqueous environments, but strong fluorescent in hydrophobic environments[64]. *E. coli* (ATCC 25922) and *P. aureginosa* (ATCC 27853) were used as representative strains for Gram-negative bacteria. Briefly, 3–5 individual colonies were inoculated into fresh tryptic soy broth (TSB) and incubated at 37 °C for 16–18 h to stationary phase. A 40 μL culture was diluted with fresh TSB by 100-fold and regrown at 37 °C to mid-log phase (OD$_{600}$ = 0.5–0.7). Bacterial cells were then harvested and washed once with sterile PBS via centrifugation (10,000 × g for 5 min at 4 °C), adjusted with sterile PBS to ~1.5 × 10$^8$ CFU/mL. A nanoparticle dispersion in sterile PBS buffer (84 μL) was added into 166 μL adjusted bacteria suspension, to achieve a final bacterial inoculum size of ~1 × 10$^8$ CFU/mL and a final nanoparticle mass concentration of 20 μg/mL. The resulting mixture was subsequently incubated at 37 °C for 3 h, treated with 1-N-Phenylnaphthylamine (NPN) in acetone (20 mM, 2.5 μL) at 37 °C for 30 min, and then subjected to fluorescence emission spectrum recording ($\lambda_{ex}/\lambda_{em}$ = 350 nm/380–500 nm, slit-widths of 5 nm and 10 nm for excitation and emission wavelengths, respectively) with a fluorimeter (F-4600 spectrofluorometer, Hitachi). Controls are NPN solution treated similarly but without nanoparticles. The reported results are averages of two independent trials.

**Assays on bacterial resistance with nanoparticles**. Whether an agent is able to delay the onset of bacterial resistance following repeated treatment is normally evaluated by performing serial passages of bacterial growth inhibition assays[40,65]. *E. coli* (ATCC 25922) and *S. aureus* (ATCC 25923) were used as representative strains for Gram-negative and -positive bacteria, respectively. Gentamicin and levofloxacin were used as two representatives for antibiotics, to help verify the reliability of our assay protocol by checking whether resistance emerges after repeated treatment with antibiotics. Briefly, 3–5 individual colonies were inoculated into fresh tryptic soy broth (TSB) (Qingdao Hope BioTechnology, China) and incubated at 37 °C for 16–18 h to stationary phase. A 40 μL culture was diluted with fresh TSB by 100-fold and regrown at 37 °C to mid-log phase (OD$_{600}$ = 0.5–0.7). For assays with antibiotics, the regrown bacterial culture was subsequently adjusted with fresh TSB to OD$_{600}$ = 0.001 (~5 × 10$^5$ cells/mL) to yield the final inoculum and serial dilutions of antibiotics were made with fresh TSB; for assays with nanoparticles, the regrown culture was subsequently adjusted with fresh MH to OD$_{600}$ = 0.001 (~5 × 10$^5$ cells/mL) to yield the final inoculum and serial dilutions of nanoparticle dispersions were made with fresh MH. Into each well of a flat-bottom 96-well microplate (Costar, Corning) were successively added an antibiotic/nanoparticle dilution (50 μL) and a final bacterial inoculum (100 μL), followed by incubation at 37 °C for 18 h. Bacterial growth was monitored by reading the optical density at 595 nm (OD$_{595}$) with a microplate reader (iMark, Bio-Rad). Controls include TSB or MH only to provide blank values for the assay readings, as well as bacterial suspensions treated similarly but without nanoparticle to indicate 100% bacterial growth. Each bacterial inhibition trial was carried out in triplicate, and the reported results are the averages of two independent trials. The reported minimum inhibition concentration (MIC) values are defined as the minimum concentrations of antibiotics or nanoparticles to inhibit 90% bacterial growth.

To assess whether repeated treatment with an antibiotic or a nanoparticle induces bacterial resistance, we performed bacterial growth inhibition assays as described above but using bacterial cultures treated at ½ MIC of a previous inhibition assay (300 μL in total, with 100 μL from each well of the triplicate) and diluted to OD$_{600}$ = 0.001 (using TBS and MH for antibiotics and nanoparticles, respectively) as the final bacterial inoculum for the next growth inhibition assay, which we called cycling inhibition assays. Such cyclic inhibition assays were continued for over 10 passages till the MIC values reached a plateau where they are steady for ≥5 consecutive assays. Each bacterial inhibition trial was carried out in triplicate, and the reported results are the averages of two independent trials.

**In vitro cytotoxicity of nanoparticles**. Murine macrophage Raw 264.7, Ana-1, murine fibroblast NIH-3T3, and murine breast cancer cell 4T1 were used as representatives for mammalian host cell lines. Briefly, approximately 10$^4$ cells were seeded into each well of a 96-well microplate, cultured in fetal bovine serum (FBS)-supplemented Dulbecco's modified Eagle's medium (DMEM) (v./v. = 10%) at 37 °C (5% CO$_2$) for ~12 h to ~80% confluency, and then treated with nanoparticles dispersion (0.1 mL in FBS-supplemented DMEM, at expected concentration) at 37 °C (5% CO$_2$) for 4 h, followed by washing with sterile PBS and replenishing with fresh FBS-supplemented DMEM (0.1 mL). The as-treated cells were subsequently incubated at 37 °C (5% CO$_2$) for another 24 h, followed by cell viability determination with a cell counting kit-8 (CCK-8) by adding 10 μL CCK-8 into each well of

a 96-well microplate and then incubating for 1 h (at 37 °C, 5% $CO_2$). Cell viability in a microplate well is indicated by the concentration of formazan therein, which is quantified by measuring the optical density at 450 nm ($OD_{450}$) with a microplate reader (Varioskan, Thermo). Cell viability ratio was defined as the relative ratio of the $OD_{450}$ of cells treated with the nanoparticle to that of cells treated similarly but without nanoparticle. Each trial was carried out in triplicate, and the reported results are averages of two independent trials.

**Cytotoxicity assays on suspended murine macrophage Ana-1 cells.** Briefly, approximately $10^4$ Ana-1 cells were seeded into each well of a 96-well microplate, cultured in FBS-supplemented RPMI 1640 Medium (v./v. = 10%) at 37 °C (5% $CO_2$) with gentle shaking (100 rpm) for ~12 h, and then treated with $AgPd_{0.38}$ dispersion (0.1 mL in FBS-supplemented RPMI 1640 Medium, at expected concentration) at 37 °C with gentle shaking (100 rpm, 5% $CO_2$) for 4 h, followed by washing with sterile PBS and replenishing with FBS-supplemented RPMI 1640 Medium (0.1 mL). The as-treated cells were subsequently incubated at 37 °C (5% $CO_2$) for another 24 h, followed by cell viability determination with a cell counting kit-8 (CCK-8) by adding 10 μL CCK-8 into each well of a 96-well microplate and then incubating for 1 h (at 37 °C, 5% $CO_2$). Cell viability in a microplate well is indicated by the concentration of formazan therein, which is quantified by measuring the optical density at 450 nm ($OD_{450}$) with a microplate reader (Varioskan, Thermo). Cell viability ratio was defined as the relative ratio of the $OD_{450}$ of cells treated with the nanoparticle to that of cells treated similarly but without nanoparticle. Each trial was carried out in triplicate, and the reported results are averages of two independent trials.

**In vitro cytotoxicity of photosensitizer-preloaded PLGA nanoparticles.** Murine macrophage Raw 264.7 was used as a representative for mammalian cell-lines. Briefly, approximately $10^4$ Raw 264.7 cells were seeded into each well of a 96-well microplate, cultured in fetal bovine serum (FBS)-supplemented Dulbecco's modified Eagle's medium (DMEM) (v./v. = 10%) at 37 °C (5% $CO_2$) for ~12 h to ~80% confluency, and then treated with nanoparticles dispersion (0.1 mL in FBS-supplemented DMEM, at expected concentration) at 37 °C (5% $CO_2$) for 4 h, followed by washing with sterile PBS and replenishing with fresh FBS-supplemented DMEM (0.1 mL). The as-treated cells were subsequently irradiated with a solar simulator (at 0.1 W/m$^2$, for 5 min). Cells were subsequently incubated at 37 °C (5% $CO_2$) for another 24 h, followed by cell viability determination with a cell counting kit-8 (CCK-8) by adding 10 μL CCK-8 into each well of a 96-well microplate and then incubating for 1 h (at 37 °C, 5% $CO_2$). Cell viability in a microplate well is indicated by the concentration of formazan therein, which is quantified by measuring the optical density at 450 nm ($OD_{450}$) with a microplate reader (Varioskan, Thermo). Cell viability ratio was defined as the relative ratio of the $OD_{450}$ of cells treated with the nanoparticle to that of cells treated similarly but without nanoparticle. Each trial was carried out in triplicate, and the reported results are averages of two independent trials.

**In vitro cellular uptake assays.** Murine macrophage Raw 264.7, Ana-1, and murine breast cancer cell 4T1 were used as representatives for mammalian host cell lines. Briefly, ~$10^5$ cells were seeded into each well of a 24-well microplate, cultured in FBS-supplemented Dulbecco's modified Eagle's medium (DMEM) (v./v. = 10%) at 37 °C (5% $CO_2$) for ~12 h to ~80% confluency, and then treated with $AgPd_{0.38}$ dispersion (0.5 mL in FBS-supplemented DMEM, at 25 μg/mL) at 37 °C (5% $CO_2$) for 4 h, followed by washing with sterile PBS. Subsequently, cells were treated with fresh aqua regia (0.5 mL) for 10 min and then diluted to a total volume of 3 mL with Millipore water. The concentrations of internalized $AgPd_{0.38}$ were measured by ICP-MS.

To understand what endocytosis pathway(s) $AgPd_{0.38}$ adopts to enter the cells, we carried out similar cellular uptake assays as described above but in the presence of an endocytosis inhibitor (amiloride, sucrose, methyl-β-cyclodextrin (M-β-CD), or dynasore). Briefly, ~$10^5$ Raw 264.7 cells were seeded into each well of a 24-well microplate, cultured in FBS-supplemented Dulbecco's modified Eagle's medium (DMEM) (v./v. = 10%) at 37 °C (5% $CO_2$) for ~12 h to ~80% confluency. Into each well of the resulting microplate was added dispersion (0.5 mL in FBS-supplemented DMEM) of an inhibitor (amiloride (2 mM), sucrose (450 mM), M-β-CD (10 mM), or dynasore (100 μM)), followed by incubation at 37 °C for 30 min, and then treated with $AgPd_{0.38}$ dispersion containing the inhibitor (25 μg/mL $AgPd_{0.38}$, 0.5 mL in FBS-supplemented DMEM) at 37 °C (5% $CO_2$) for 4 h, and then washed with sterile PBS, to reveal how the presence of an endocytosis inhibitor impacts the cellular uptake of $AgPd_{0.38}$. To indicate whether the internalization of $AgPd_{0.38}$ is energy-dependent, the overnight cell culture was incubated at 4 °C for 30 min, followed by the addition of $AgPd_{0.38}$ dispersion (25 μg/mL in FBS-supplemented DMEM, 0.5 mL) and then incubation at 4 °C for 4 h. Subsequently, cells were treated with fresh aqua regia (0.5 mL) for 10 min and then diluted to a total volume of 3 mL with Millipore water. The concentrations of internalized $AgPd_{0.38}$ were measured by ICP-MS. Each trial was carried out in triplicate, and the reported results are averages of two independent trials.

**Cellular uptake assays on suspended murine macrophage Ana-1 cells.** Briefly, approximately $10^5$ Ana-1 cells were seeded into each well of a 24-well microplate,

cultured in FBS-supplemented RPMI 1640 Medium (v./v. = 10%) at 37 °C (5% $CO_2$) with gentle shaking (100 rpm) for ~12 h, and then treated with $AgPd_{0.38}$ dispersion (0.5 mL in FBS-supplemented RPMI 1640 Medium, at 25 μg/mL) at 37 °C (5% $CO_2$) for 4 h, followed by washing with sterile PBS. Subsequently, cells were treated with fresh aqua regia (0.5 mL) for 10 min and then diluted to a total volume of 3 mL with Millipore water. The concentrations of internalized $AgPd_{0.38}$ were measured by ICP-MS.

**In vitro cytotoxicity of $AgPd_{0.38}$ to cells in the presence of endocytosis inhibitor.** To understand how endocytosis affects the cytotoxicity of $AgPd_{0.38}$, we carried out similar in vitro cytotoxicity CCK-8 assays as described above but in the presence of an endocytosis inhibitor (dynasore), using Raw 264.7 as a representative for mammalian cell-lines. Briefly, approximately $10^4$ cells were seeded into each well of a 96-well microplate, cultured in fetal bovine serum (FBS)-supplemented Dulbecco's modified Eagle's medium (DMEM) (v./v. = 10%) at 37 °C (5% $CO_2$) for ~12 h to ~80% confluency. Into the resulting cell culture was added sterile dynasore dispersion (100 μM in FBS-supplemented DMEM, 0.1 mL), followed by incubation at 37 °C (5% $CO_2$) or 4 °C for 30 min, and then treated with $AgPd_{0.38}$ dispersion (0.1 mL in FBS-supplemented DMEM containing 100 μM dynasore, at expected concentration) at 37 °C (5% $CO_2$) or 4 °C for 4 h, followed by washing with sterile PBS and replenishing with fresh FBS-supplemented DMEM (0.1 mL). The as-treated cells were subsequently incubated at 37 °C (5% $CO_2$) for another 24 h, followed by cell viability determination with a cell counting kit-8 (CCK-8) by adding 10 μL CCK-8 into each well of a 96-well microplate and then incubating for 1 h (at 37 °C, 5% $CO_2$). Cell viability in a microplate well is indicated by the concentration of formazan therein, which is quantified by measuring the optical density at 450 nm ($OD_{450}$) with a microplate reader (Varioskan, Thermo). Cell viability ratio was defined as the relative ratio of the $OD_{450}$ of cells treated with the nanoparticle to that of cells treated similarly but without nanoparticle. Each trial was carried out in triplicate, and the reported results are averages of two independent trials.

**Performance of nanocages in mouse models.** To assess the in vivo antibacterial performance and potential safety of AgPd nanocages, we established a skin wound-infected mouse model, using *P. aureginosa* (ATCC 27853) as the pathogen. All animal experiments were conducted in compliance with the guidelines for the care and use of research animals established by the Animal Care and Use Committee at the University of Science and Technology of China (USTCACUC1501010). Briefly, twelve ICR female mice (6–8-week old) were obtained from Beijing Vital River Laboratory Animal Technology Co., Ltd, and their backs were cut to generate a circular wound (diameter ~0.5 cm) and then challenged with *P. aureginosa* (~$1 \times 10^8$ CFU/mL, 50 μL dispersed in PBS) to generate the infected wounds. At 48 h after infection, the mice were randomly divided into 2 groups (n = 6 in each group), and treated with $AgPd_{0.38}$ or PBS, respectively. On 1-, 3-, 9-days after treatment, three mice of each group (n = 3) were randomly selected for blood collection (~100 μL), and the as-collected blood samples were assayed with an ELISA kit (Beyotime, China) to determine the serum levels of IL-1β, IL-6, and TNF-α (three representatives for pro-inflammation cytokines). Controls include blood samples collected one day prior to bacterial infection (i.e., day −3) to indicate the serum levels of above three cytokines in healthy mice and those collected one day prior to treatment initiation (i.e., day −1) to indicate the serum levels of above three cytokines in infected yet untreated mice.

The wounds of infection were photographed every other day throughout the whole observation window. On the 9th day after treatment initiation (i.e., day 9), all mice were sacrificed and their wound tissues and major organs (which are hearts, livers, spleens, lungs, and kidneys) were collected. The wound tissues from each group were randomly divided into two sub-groups, with one sub-group (n = 4) for bacterial colony forming units (CFU) determination and another (n = 2) for hematoxylin and eosin (H&E) tissue staining effect analysis. To determine the number of bacteria in the wound tissue, the wound tissues from four randomly selected mice from each group (n = 4) were placed into sterile PBS (0.2 mL) and then homogenized using a homogenizer (Precellys Evolution, Bertin Instruments) for 2 min, followed by 10-fold serial dilution with sterile PBS. 20 μL of the resulting dilutions were plated onto TSB agar plates, followed by 18-h incubation at 37 °C to form visible colonies. For histological analysis, the wound tissues (n = 2) and major organs (n = 6) from each group of mice were fixed in 10% neutral buffered formalin, processed routinely into paraffin, sectioned, stained with hematoxylin and eosin for H&E analysis.

**Modifying surface with $AgPd_{0.38}$.** The poly(dimethylsiloxane) (PDMS) silicone was used as the model for biomedical devices and prepared following the standard protocol of SYLGARD 184 elastomer kit (Dow Corning, USA). Briefly, the monomer (15 mL) was mixed with a curing agent (1.5 mL) at a v./v. the ratio of 10:1 and cured in a plastic Petri dish (d ~90 mm) at 80 °C for 12 h. Subsequently, the silicon wafers were soaked in hexane for 12 h to remove the unreacted reagents, and then dried at 37 °C for 12 h to remove residual hexane. The as-prepared silicone wafers were cut into small circular ones with a diameter of 1 cm, which were subsequently coated successively with polydopamine (PDA)[66] and $AgPd_{0.38}$, leading to surfaces named as PDMS/PDA and PDMS/PDA/AgPd, respectively. Briefly, the circular silicon wafers (n = 6) were added into a beaker, followed by

successive additions of Millipore water (149 mL), dopamine hydrochloride (300 mg), and Tris-HCl buffer (1.5 M Tris-HCl at pH = 8.8, 1 mL) and then gentle stirring (200 rpm) at ~15 °C for 24 h. The as-treated silicone wafers were subsequently taken out, washed with Millipore water once, and then dried at 37 °C for 4 h, which yielded the polydopamine (PDA)-coated PDMS (PDMS/PDA) wafers.

AgPd$_{0.38}$ was impregnated onto the PDMS/PDA wafer by following a previously reported procedure[66,67]. Briefly, a PDMS/PDA wafer was immersed into a AgPd$_{0.38}$ dispersion (0.2 mg/mL, 1 mL, pH = 3.6), followed by gentle vibration (~100 rpm) for 24 h. The as-treated wafer was subsequently taken out, washed with Millipore water once, and then dried at 37 °C for 4 h, which yielded the expected AgPd$_{0.38}$-coated wafer, PDMS/PDA/AgPd.

Fourier-transform infrared spectroscopy (FT-IR) spectra of the pristine PDMS and PDMS/PDA wafers were recorded with an FT-IR spectrophotometer (TENSOR27, BRUKER) and compared to verify the successful attachment of PDA onto the PDMS wafer. The water contact angles of the pristine PDMS, PDMS/PDA, and PDMS/PDA/AgPd wafers were measured with a contact angle analyzer (SL2003, Solon Tech. (Shanghai) Co., Ltd) and compared to confirm the successful attachment of PDA onto the PDMS wafer and successful impregnation of AgPd$_{0.38}$ onto the PDMS/PDA wafer. Surface morphology of the pristine PDMS, PDMS/PDA, and PDMS/PDA/AgPd wafers were imaged with a scanning electron microscope (SEM) (FEI Apreo, USA), to confirm the successful attachment of PDA onto the PDMS wafer and successful impregnation of AgPd$_{0.38}$ onto the PDMS/PDA wafer. To determine the amount of AgPd$_{0.38}$ impregnated onto a PDMS/PDA wafer, we added a PDMS/PDA/AgPd wafer into aqua regia (1 mL), and the resulting solution was then measured with a Thermo Scientific PlasmaQuad 3 inductively coupled plasma mass spectrometry (ICP-MS) to quantify the amount of Pd and Ag therein.

### In vitro protein adsorption assays
Protein adsorption onto the surfaces was evaluated through a previously reported procedure. Briefly, PDMS/PDA/AgPd wafer was washed with PBS once and then placed onto the bottom of a well in a 24-well plate, followed by the addition of bovine serum albumin (BSA) solution (2 mg/mL in H$_2$O, 1 mL) to incubated at 37 °C for 5 days. The as-treated wafer was subsequently washed successively with PBS once and Millipore water twice and then was immersed into Millipore water (1 mL) containing sodium dodecyl sulfate (SDS) (2.0 wt.%), followed by shaking (200 rpm) for 2 h and then sonication for 15 min at room temperature to remove the surface-attached BSA. The resulting supernatant was collected for BSA concentration determination with a BCA kit, which measures the absorbance at 562 nm using a microplate reader (Varioskan, Thermo). Controls include the pristine PDMS and PDMS/PDA wafers treated under comparable conditions. The reported results are averages of two independent trials.

### Bactericidal activity of PDMS/PDA/AgPd wafers
We assessed the bactericidal activity of PDMS/PDA/AgPd wafers by performing plate bacterial killing assays. *E. coli* (ATCC 25922) and *P. aureginosa* (ATCC 27853) were used as representative strains for Gram-negative bacteria, while *S. aureus* (ATCC 25923) and *B. subtilis* (ATCC 6051) stains were used as representative Gram-positive bacteria. Briefly, 3–5 individual colonies were inoculated into fresh tryptic soy broth (TSB) and incubated at 37 °C for 16–18 h to stationary phase. A 40 μL culture was diluted with fresh TSB by 100-fold and regrown at 37 °C to mid-log phase (OD$_{600}$ = 0.5–0.7). Bacterial cells were then harvested and washed once with sterile PBS via centrifugation (10,000 × $g$ for 5 min at 4 °C), adjusted with sterile PBS to ~1.5 × 10$^6$ CFU/mL. 100 μL adjusted bacteria suspension was dropped on the surface of a PDMS/PDA/AgPd wafer in a Petri dish ($d$ ~ 90 mm), followed by incubation at 37 °C for 3 h. The wafer was subsequently transferred into sterile PBS (5 mL), followed by mixing via vibration with a lab dancer (IKA). Serial dilutions were subsequently made with sterile PBS buffer and the dilutions (20 μL) were then plated onto TSB agar plates, followed by overnight incubation at 37 °C to form visible colonies. Controls include bacterial culture treated similarly but with the pristine PDMS wafer to indicate the inoculum size and that treated similarly but with the PDMS/PDA wafer to confirm that the bactericidal activity of the PDMS/PDA/AgPd wafer arises because of AgPd$_{0.38}$ impregnation. The reported results are the averages of two independent trials.

### In vitro antibiofilm assays
*P. aeruginosa* (PAO1) (ATCC BAA-47), which naturally expresses green fluorescent protein (GFP)[58], was used as a representative strain for biofilm-forming bacteria. Briefly, 3–5 individual colonies were inoculated into fresh tryptic soy broth (TSB) and incubated at 37 °C for 16–18 h to stationary phase. A 40 μL culture was diluted with fresh TSB by 100-fold and regrown at 37 °C to mid-log phase (OD$_{600}$ = 0.5–0.7). Bacterial cells were then harvested and washed once with sterile PBS via centrifugation (10,000 × $g$ for 5 min at 4 °C), adjusted with sterile PBS to ~1 × 10$^8$ CFU/mL. A PDMS/PDA/AgPd wafer was placed onto the bottom of a well in a 24-well plate, followed by the addition of PAO1 dispersion (~1 × 10$^8$ CFU/mL, 1 mL) and then culture at 37 °C for 5 days (replenishing with fresh TSB on the 3rd day) to allow biofilm formation.

The PDMS/PDA/AgPd wafer was taken out from the culture once the 5-day culture was completed, washed with sterile PBS once, and then transferred onto a coverslip, air-dried, immersed in PBS (~20 μL), and imaged under fluorescence

microscopy (IX81, Olympus) using a ×100 oil-immersion objective lens. For fluorescence confocal microscope (SP5, Leica) imaging, PAO1 was visualized by using an excitation laser line of 488 nm (intensity: 1.2%) and an emission wavelength range of 490–600 nm. The same wafer was then fixed with 4% paraformaldehyde (2 mL) for 1 h, dehydrated successively with a series of graded ethanol solutions (25, 50, 75, 90, and 100%) (2 mL), cut into a suitable size for being subsequently mounted onto a conductive tape, dried overnight, sputtered with gold for 30 s, and then imaged under a scanning electron microscope (SEM) (Apreo, FEI).

Controls include bacterial culture treated similarly but with the pristine PDMS wafer to indicate the inoculum size and that treated similarly but with the PDMS/PDA wafer to confirm that the bactericidal and antibiofilm activity of the PDMS/PDA/AgPd wafer arises because of AgPd$_{0.38}$ impregnation.

### Coating silicone catheters with AgPd$_{0.38}$
AgPd$_{0.38}$ was impregnated onto silicone catheters through a protocol as described above in "Modifying a surface with AgPd$_{0.38}$" section. Briefly, a sterile silicone catheter (inner diameter = 2.2 mm, outer diameter = 3.4 mm) was cut into sections of 4.5 cm in length. The silicone catheters ($n$ = 8) were added into a beaker, followed by successive additions of Millipore water (149 mL), dopamine hydrochloride (300 mg), and Tris-HCl buffer (1.5 M Tris-HCl at pH = 8.8, 1 mL) and then gentle stirring (200 rpm) at ~15 °C for 24 h. The as-treated silicone catheters were subsequently taken out, washed with Millipore water once, and then dried at 37 °C for 4 h, which yielded the polydopamine (PDA)-coated catheters (Ca/PDA). A resulting Ca/PDA catheter was then immersed into a AgPd$_{0.38}$ dispersion (0.2 mg/mL, 1 mL, pH = 3.6), followed by gentle vibration (~100 rpm) for 48 h. The as-treated catheter was subsequently taken out, washed with Millipore water once, and then dried at 37 °C for 4 h, which yielded the expected AgPd$_{0.38}$-coated catheter, Ca/PDA/AgPd.

To determine the amount of AgPd$_{0.38}$ impregnated onto a PDMS/PDA wafer, we added a Ca/PDA/AgPd catheter into aqua regia (1 mL), and the resulting solution was then measured with a Thermo Scientific PlasmaQuad 3 inductively coupled plasma mass spectrometry (ICP-MS) to quantify the amount of Pd and Ag therein.

### Performance of Ca/PDA/AgPd catheter in mouse models
To evaluate the antibacterial effect and potential safety of Ca/PDA/AgPd Catheter in vivo, we established a mouse skin wound model[68]. All animal experiments were conducted in compliance with the guidelines for the care and use of research animals established by the Animal Care and Use Committee at the University of Science and Technology of China. PAO1 was used as a representative for an infective pathogen. Fifteen ICR female mice (6–8-week old) were obtained from Beijing Vital River Laboratory Animal Technology Co., Ltd. Their backs were cut to generate a circular wound (diameter ~ 0.6 cm) and challenged with PAO1 culture (~1 × 10$^9$ CFU/mL, 50 μL dispersed in PBS). At 48-h after infection, these mice were randomly divided into 3 groups ($n$ = 5 per group), which were subsequently treated with PBS (20 μL) (i.e., control), Ca/PDA, and Ca/PDA/AgPd, respectively, for 5 days. The wounds were photographed and measured every day throughout the whole observation window. The mouse weights were recorded every day throughout the whole observation window. On the 5th day after treatment initiation (i.e., day 5), all Ca/PDA and Ca/PDA/AgPd were removed from the wounds, collected, and cut into halves along the cross-section with one half for imaging under fluorescence microscopy. To view whether biofilm was formed, the as-halved catheters were further cut along the cylinder axis and imaged under fluorescence microscopy (IX81, Olympus) using a ×100 oil-immersion objective lens.

At 1-, 3-, 8- days after treatment, blood samples were collected (~100 μL) from each mouse for blood routine test using hematology analyzer (XT-1800i, Sysmex). Controls include blood samples collected one day prior to bacterial infection (i.e., day −3) to indicate the blood cell level in healthy mice and those collected one day prior to treatment initiation (i.e., day −1) to indicate the blood cell level in infected yet untreated mice.

On the 8th day after treatment initiation (i.e., day 8), all mice were sacrificed and their wound tissues were collected. The wound tissues from each group were randomly divided into two sub-groups, with one sub-group ($n$ = 2) for hematoxylin and eosin (H&E) tissue staining effect analysis and another ($n$ = 3 for the Ca/PDA/AgPd group, $n$ = 2 for the control of Ca/PDA groups due to mouse death during the observation window) for CFU determination. The colony-forming units (CFUs) in a wound were determined via a plating method. Briefly, to determine the CFU in the wound tissue, the wound tissues from randomly selected mice from each group were placed into sterile PBS (0.2 mL) and then homogenized using a homogenizer (Precellys Evolution, Bertin Instruments) for 2 min, followed by 10-fold serial dilution with sterile PBS. 20 μL of the resulting dilutions were plated onto TSB agar plates, followed by 18-h incubation at 37 °C to form visible colonies. For histological analysis, the wound tissues ($n$ = 2) were fixed in 10% neutral buffered formalin, processed routinely into paraffin, sectioned, stained with hematoxylin and eosin for H&E analysis.

### Statistical analysis
All the statistical analyses were performed using Origin 8.0, Excel 2013, and Nano Measurer software. Statistical comparisons were carried out

by performing a two-sided Student's $t$-test. *, ** and *** indicate $p < 0.05$, $p < 0.01$, and $p < 0.001$, respectively.

**Reporting summary**. Further information on research design is available in the Nature Research Reporting Summary linked to this article.

## Data availability

The authors declare that data supporting the findings of this study are available within the paper and its Supplementary Information files. Source data are provided with this paper.

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

## Acknowledgements

We gratefully thank Professor Yucai Wang at USTC for use of his facilities. This work was supported in part by the National Key R&D Program of China (2017YFA0207301), the National Natural Science Foundation of China (31671014, 21725102, 91961106), and the Ministry of Education of China (through the Fundamental Research Funds for the Central Universities in China) (WK3450000005). The synchrotron soft X-ray microscopy experiments were carried out at the National Synchrotron Radiation Laboratory at the University of Science and Technology of China (beamline BL07W).

## Author contributions

L.Y. and Y.X. conceived the idea; T.S. and Y.X. contributed the nanocages; F.G., Y.Y. and T.S. performed the experiments; L.Y., F.G. and T.S. analyzed the results; L.Y., F.G. and Y.X. wrote the paper.

## Competing interests

A patent application was submitted to China National Intellectual Property Administration (patent applicant: University of Science and Technology of China; names of inventors: Lihua Yang, Yujie Xiong, Feng Gao, Tianyi Shao, Ming Li; application number: 202010847666.6; status of application: submitted; specific aspect of manuscript covered in the patent application: the preparation of AgPd nanocages and characterizations using transmission electron microscopy and Inductively Coupled Plasma Mass Spectrometry, colloidal stability of AgPd nanocages in aqueous solutions, ROS production by the nanocages using ascorbic acid as the ROS probe, in vitro plate-killing assays against bacteria, serial bacterial inhibition assays to examine whether AgPd nanocages induce resistance after repeated treatment, in vitro cytotoxicity assays to murine macrophage and fibroblast cells, tests on wound disinfection and healing in mouse models bearing bacterial-infected skin wounds).
