## [Peer Review File · Nature Communications]

REVIEWER COMMENTS

Reviewer #1 (Remarks to the Author):

In this paper the authors developed an antibacterial strategy using oxidase-like AgPd nanozymes to generate surface-bound ROS which can preferentially kill bacteria. Such antibacterial mechanism is promising in tackling antibacterial resistance (AMR) in encoded species or biofilm. The devices of PDMS or catheter coated with AgPd nanozymes showed impressive antibacterial efficacy both in vitro and in vivo for biofilm prevention and wound healing. Thus I think it is valuable for consideration to be accepted for publication on Nature Communications.

Below are some concerns and questions.

1. Right now the name of nanozymes is the most frequently used word to describe nanomaterials with enzyme-like activities. Please change all nanoenzymes into nanozymes in the manuscript.
2. For AgPd nanocages, is Pd exist on the outer surface and Ag located in the center? Does Ag contribute to catalysis and bacteria killing?
3. Since AgPd nanocages perform oxidase-like activity, what is the optical reaction condition including pH, buffer, temperature, substrate?
4. To prove surface-bound ROS, the authors introduced a lipid layer to cover AgPd_{0.38}. However, lipid peroxidation may be induced by ROS in lipid layer (it can be identified with MDA index). So such evidence is not convincing to ensure ROS only bound on nano-surface.
5. To confirm ROS not freely dispersing, a transwell experiment or immobilizing AgPd nanocages on a solid surface can be used to check if it can kill bacteria or not. I suspect the conclusion that ROS is only bound on AgPd surface and cannot freely disperse. If it is true, the antibacterial activity will be severely reduced under physiological condition in which many biomolecules and cells may attach on AgPd surface and block the interaction with bacteria. However, according to the in vivo experiments, AgPd nanocages still showed considerable bacterial killing. Especially when it is immobilized on catheter, CA/PDA/AgPd still can promote wound healing.
6. In Figure S17, SYTO-9 can stain both live and dead bacteria.
7. The style of references need to be checked carefully.

Reviewer #2 (Remarks to the Author):

In this manuscript, Gao et al. synthesized AgPd nanoenzymes for the treatment against bacterial infection, among which AgPd_{0.38} seems to have the best antibacterial efficacy against different bacteria strains. By generating surface-bound ROS, the nanoenzyme showed preferential killing of bacteria over mammalian cells in vitro and promoted wound disinfection in mouse models. The AgPd_{0.38} nanoenzyme also showed ability to kill antibiotic-resistant bacteria, inhibit biofilm formation, and suppress immune response in vivo. Overall, the reported nanoenzyme shows promising antimicrobial activity and may have good applications. However, the current version of the manuscript lacks fundamental understandings of the system. There is little knowledge on the preferential killing to bacteria over cells; the proposed mechanism and associated experimental proofs demonstrate inadequate support. Below are some detailed comments that may help the authors to further improve the manuscript.

1. The title of the manuscript can be improved to better reflect the content of the manuscript. The authors aimed for a big theme, but only completed a small part. Particularly, neither preferential

killing nor surface-binding have been clearly demonstrated throughout the manuscript.

2. Reduced efficiency from lipid-coated AgPd doesn't necessarily mean the surface-binding of ROS, which could be from other factors such as fast ROS reacting with lipid, slow diffusion of the substrate, etc. It's not convincing to attribute the reason of reduced efficiency to the surface-binding without ruling out other possibilities.

3. The authors claimed the endocytosis of AgPd by mammalian cells may confine the cytotoxicity of the ROS within endocytic vesicles, despite only 27% of AgPd could be taken up by the co-incubated cells. Is it possible that mammalian cells are less sensitive to the ROS generated by AgPd than the bacteria? Therefore, no matter where ROS is located, it only has modest damage to the cells.

4. Higher IC50 in cells than those in bacteria doesn't necessarily mean the preferential killing, again other factors need to be carefully considered. Presumably, a similar nanoenzyme without any endocytosis could present similar effect. The higher sensitivity of bacteria to AgPd doesn't mean the off-target to cells has been reduced (Line 328).

5. The authors attempted to highlight the biomedical benefits of preferential killing, however without any appropriate controls, it becomes less convincing. Lipid-coated AgPd could be used as a negative control presumably with reduced bacterial killing, while another potent nanoenzyme with no endocytosis could be used as a control to emphasize the necessity of reduced cell uptake for enhanced biosafety of nanoenzyme.

6. Besides plots, other experimental methods are much anticipated, such as microscopic imaging, protein blots, ROS sensing kinetics, etc. Moreover, a schematic illustration could better support the explanations.

Responses to Reviewer Comments

What nanozymes preferentially kill bacteria over mammalian cells and bio-compatibly inhibit biofilm formation (those generating surface-bound reactive oxygen species)

Feng Gao[#], Tianyi Shao[#], Yunpeng Yu, Yujie Xiong, and Lihua Yang**

~~~~~

### ***Response to Reviewer #1:***

*In this paper the authors developed an antibacterial strategy using oxidase-like AgPd nanozymes to generate surface-bound ROS which can preferentially kill bacteria. Such antibacterial mechanism is promising in tackling antibacterial resistance (AMR) in encoded species or biofilm. The devices of PDMS or catheter coated with AgPd nanozymes showed impressive antibacterial efficacy both in vitro and in vivo for biofilm prevention and wound healing. Thus I think it is valuable for consideration to be accepted for publication on Nature Communications.*

Author Response: We gratefully thank the reviewer for the positive remarks.

*Below are some concerns and questions.*

*1. Right now the name of nanozymes is the most frequently used word to describe nanomaterials with enzyme-like activities. Please change all nanoenzymes into nanozymes in the manuscript.*

Author Response: We thank the reviewer for the kind suggestion. In revision, all nanoenzymes have been accordingly changed to nanozymes.

*2. For AgPd nanocages, is Pd exist on the outer surface and Ag located in the center? Does Ag contribute to catalysis and bacteria killing?*

Author Response: We thank the reviewer for the critical comments. In fact, the AgPd nanocages are not in a core-shell structure (with Pd on the outer surface

while Ag in the center). Instead, using  $\text{AgPd}_{0.38}$  as a representative for these nanocages, we found that  $\text{AgPd}_{0.38}$  has a hollow interior and a surface composed of both Pd and Ag (Figure S6a). This was confirmed with energy dispersive spectroscopy images of  $\text{AgPd}_{0.38}$  (Figure S6b), which we characterized in revision. Moreover, high-resolution TEM images of  $\text{AgPd}_{0.38}$  (Figure S6c) further show that, in a single  $\text{AgPd}_{0.38}$  particle, there co-exists AgPd alloy grains, pure Pd grains, and pure Ag grain.

**Figure S6.** (a) Elemental mapping profiles of  $\text{AgPd}_{0.38}$  under energy dispersive spectroscopy (EDS), in which red and green signals indicate Pd and Ag elements, respectively. (b) EDS line scan profiles of an individual  $\text{AgPd}_{0.38}$  particle, in which red and green lines indicate Pd and Ag signals, respectively. (c) High resolution TEM images showing two selected areas of  $\text{AgPd}_{0.38}$ .

And we think Ag in AgPd nanocages contributes to catalysis and bacterial killing, for reasons as follows. For these AgPd nanocages, there exists a non-monotonic relationship between ROS production *versus* Pd content (Figure 1b and Figure S4-5) and same is true between antibacterial activity *versus* Pd content (Figure 2a and Figure S21), despite that the precursor Ag nanocube (AgNC) (Figure S2) was inefficient in catalyzing ROS production and inactive against bacteria (Figure S26). If Ag in AgPd nanocages did not contribute to catalysis or bacterial killing, increase in Pd content (*i.e.*, decrease in Ag content) should have resulted in monotonic enhancement in ROS production and in antibacterial activity, which clearly contradicts with our experimental observations. Therefore, Ag in AgPd nanocages does contribute to catalysis and bacterial killing.

As to how Ag in AgPd nanocages contributes to catalysis and bacterial killing, our guess is: the partially alloyed structure of AgPd nanocages, as exemplified with  $\text{AgPd}_{0.38}$  (Figure S6c), indicates that Ag in the precursor AgNC used for preparing AgPd nanocages through the galvanic replacement method (*Nano Res.* 2016, 9, 1590-1599) forms AgPd alloy grains with infiltrating Pd as well as exists as residue pure Ag grains. In such a partially alloyed structure, the strain due to lattice mismatch at the grain interfaces (AgPd alloy *versus* Ag *versus* Pd) may control the catalyzing activity, as observed previously with the partially de-

alloyed fuel cell catalysts (*Nat. Chem.* 2010, 2, 454-460). And higher catalyzing activity (*i.e.*, more ROS production) corresponds to stronger bacterial killing activity (Figure S21). Nevertheless, it should be noted that, though silver ion ( $\text{Ag}^+$ ) is known to be antibacterial (*ACS Nano* 2014, 8, 374-386; *ACS Appl. Mater. Interfaces* 2018, 10, 8443–8450; *Appl. Environ. Microbiol.* 2008, 74, 2171-2178), the possibility that AgPd nanocages kill bacteria by releasing  $\text{Ag}^+$  is excluded, because the Ag leakage from AgPd nanocages into the environmental solution is negligible (Figure S13).

**Figure S13.** (a) The concentrations of Pd and Ag elements in the environmental solutions and (b) their relative percentages to the initial amount of  $\text{AgPd}_{0.38}$  after 3- and 18-h incubation of  $\text{AgPd}_{0.38}$  ( $40 \mu\text{g/mL}$ ) in PBS at  $37^\circ\text{C}$ . # in (a) indicates concentrations  $\leq 0.08 \mu\text{g/mL}$ . # in (b) indicates percentage of  $\leq 0.2\%$ . Data points are reported as mean  $\pm$  standard deviation ( $n=2$ ).

3. Since AgPd nanocages perform oxidase-like activity, what is the optical reaction condition including pH, buffer, temperature, substrate?

Author Response: We thank the reviewer for the suggestion. In revision, we have accordingly examined how reaction conditions, specifically pH, buffer, temperature, and environmental atmosphere, affect the ROS production by AgPd nanocages, using  $\text{AgPd}_{0.38}$  as the representative.

To examine the effects of pH on  $\text{AgPd}_{0.38}$ 's ROS production, we used ABDA as the ROS probe, as it — in contrast to AA and SOSG — retains its function as an ROS probe throughout the whole examined pH range (pH of 1-12) (Figure S8a-c), and monitored the ROS production by comparing the fluorescence emission spectra of ABDA at a specific pH in the presence *versus* absence of  $\text{AgPd}_{0.38}$  (Figure S8d-o). Our results revealed that pH exerts negligible effects on the ROS production by  $\text{AgPd}_{0.38}$  (Figure S8p).

**Figure S8.** (a) Absorption spectra of AA and (b-c) fluorescence emission spectra of (b) ABDA and (c) SOSG after 3-h incubation in PBS at different pH. (d-o) Fluorescence emission spectra of ABDA after 3-h treatment with AgPd0.38 (64  $\mu\text{g}/\text{mL}$ ) in PBS at different pH, with those of ABDA treated similarly but with PBS included as controls. (p) Relative fluorescence intensity of AgPd0.38-treated ABDA to its corresponding PBS-treated counterpart. Data points are reported as mean  $\pm$  standard deviation ( $n = 2$ ).

To examine the effects of temperature on  $\text{AgPd}_{0.38}$ 's ROS production, we used ABDA and SOSG as the ROS probes, as they — in contrast to AA — function as ROS probes in the examined temperature range (4-55 °C) almost in a temperature-insensitive manner (Figure S9a-c), and monitored the ROS production by comparing the fluorescence emission spectra of ABDA (Figure S9) or SOSG (Figure S10) in PBS at pH 7.4 in the presence *versus* absence of  $\text{AgPd}_{0.38}$ . Our results revealed that environmental temperature exerts negligible effects on the ROS production by  $\text{AgPd}_{0.38}$  (Figure S9j and Figure S10g).

**Figure S9.** (a) Absorption spectra of AA and (b-c) fluorescence emission spectra of ABDA and (c) SOSG after 3-h incubation in PBS at different temperature. (d-i) Fluorescence emission spectra of ABDA after 3-h treatment with  $\text{AgPd}_{0.38}$  (64  $\mu\text{g}/\text{mL}$ ) in PBS at different temperature, with those of ABDA treated similarly but with PBS included as controls. (j) Relative

fluorescence intensity of AgPd0.38-treated ABDA to its corresponding PBS-treated counterpart. Data points are reported as mean ± standard deviation (n = 2).

**Figure S10.** (a-f) Fluorescence emission spectra of SOSG after 3-h treatment with AgPd0.38 (64  $\mu\text{g}/\text{mL}$ ) in PBS at different temperature, with those of SOSG treated similarly but with PBS included as controls. (g) Relative fluorescence intensity of AgPd0.38-treated SOSG to its corresponding PBS-treated counterpart. Data points are reported as mean ± standard deviation (n = 2).

To examine the effects of buffer on AgPd0.38's ROS production, we used four different buffers (which are PBS, Tris-HCl, HEPES, and Hanks) and monitored the ROS production at pH 7.4 and 37 °C by comparing the absorption spectra of AA or fluorescence emission spectra of ABDA in a buffer in the presence *versus* absence of AgPd0.38 (Figure S11 a-h). Our results revealed that changing the buffer agents brings about negligible effects on the ROS production by AgPd0.38 (Figure S11 i).

**Figure S11.** (a-d) Absorption spectra of AA after 3-h treatment with AgPd0.38 (8  $\mu\text{g}/\text{mL}$ ) in different buffers, with those of AA treated similarly but with the corresponding buffers included as controls. (e-g) Fluorescence emission spectra of ABDA after 3-h treatment with AgPd0.38 (64  $\mu\text{g}/\text{mL}$ ) in different buffers, with those of ABDA treated similarly but with the corresponding buffers included as controls. Buffers used here include PBS, Tris-HCl, HEPES, and Hanks, and the pH of these buffers was set to be constant at 7.4. (i) Relative fluorescence intensity of AgPd0.38-treated ABDA to its corresponding buffer-treated counterpart. Data points are reported as mean  $\pm$  standard deviation ( $n = 2$ ).

To examine the effects of substrate on AgPd0.38's ROS production, we carried out two sets of experiments. In the original manuscript, we showed that removing dissolved oxygen (*via* N2 purging for 30 min) from nanocage dispersion and AA solution prior to their mixing significantly reduced AgPd0.38's ability to reduce the absorbance of AA at 266 nm, indicative of retarded production of ROS due to oxygen removal (Figure 1c in the original manuscript), suggesting oxygen as the substrate for AgPd0.38 to generate ROS. In revision, we carried out similar assays but removing dissolved O2 *via* Ar purging (also for 30 min) and observed

similarly retarded ability of AgPd0.38 to reduce AA's absorption (at 266 nm) due to O2 removal (Figure 1c and Figure S12). Collectively, these results suggest O2, rather than Ar or N2, as the substrate for our oxidase-like nanocages.

**Figure S12.** Absorption spectra of AA after 3-h treatment with AgPd0.38 (8 μg/mL) in PBS purged with (a) N2, (b) Ar and (c) O2. Those of AA treated similarly but without AgPd0.38 (*i.e.*, with PBS) were included as controls.

4. To prove surface-bound ROS, the authors introduced a lipid layer to cover AgPd0.38. However, lipid peroxidation may be induced by ROS in lipid layer (it can be identified with MDA index). So such evidence is not convincing to ensure ROS only bound on nano-surface.

Author Response: We thank the reviewer for the critical comment and inspiring suggestion. We agree with the reviewer that lipid peroxidation may be induced by the ROS, especially considering the significant content of DOPC, a lipid with unsaturated carbon-carbon double bond, in the lipid bilayer (DOPC : DSPE-PEG = 0.90 : 0.10, in weight ratio) we used to coat nanoparticles.

To assess whether the outward translocation of ROS is appreciably affected by the lipid bilayer coating (*e.g.*, as a physical barrier, *via* consumption of ROS), we used Chlorin e6 (Ce6)-preloaded PLGA (poly(lactic-co-glycolic acid)) nanoparticle (*i.e.*, Ce6/PLGA)—Ce6 is a photodynamic sensitizer that generates free ROS upon irradiation with near-infrared light ( $\lambda \sim 660$  nm)—as a model for free-ROS generating nanoparticles and examined how coating Ce6/PLGA with a lipid bilayer of same composition (DOPC : DSPE-PEG = 0.90 : 0.10) affects its ability to oxidize SOSG (an ROS probe) in the bulk solution upon light irradiation. Our results (Figure 1h) show that the fluorescence emission spectrum of SOSG treated with the as-coated nanoparticle, Ce6/PLGA@lipid, was nearly identical to that of SOSG treated similarly but with Ce6/PLGA, indicative of un-affected oxidation of SOSG in the bulk solution by the free ROS generated by PLGA-encapsulated Ce6 upon light irradiation, suggesting that the lipid bilayer coating imposes negligible effects on the outward translocation of free ROS (*e.g.*, free 1O2).

**Figure 1.** (f) Schematic illustration on the preparation of Ce6/PLGA@lipid, done by coating a Chlorin e6 (Ce6)-preloaded PLGA (poly(lactic-co-glycolic acid)) nanoparticle (*i.e.*, Ce6/PLGA) with a PEGylated lipid bilayer. (g) TEM images of Ce6/PLGA and Ce6/PLGA@lipid. Red arrow indicates the lipid bilayer coating. (h) Fluorescence emission spectra of SOSG treated with Ce6/PLGA@lipid or Ce6/PLGA ( $5 \mu\text{g/mL}$  in Ce6 dose) upon irradiation with a solar simulator (at  $0.1 \text{ W/m}^2$ , 5 min), with that of SOSG treated similarly but with PBS included for as a control. (i) Schematic illustration on the inability of the lipid bilayer coating in Ce6/PLGA@lipid to affect the outward translocation of free ROS generated by PLGA-encapsulated Ce6 upon light irradiation, thereby leading to unaffected oxidation of ROS probes in the bulk solution compared to Ce6/PLGA.

In revision, we examined how coating Ce6/PLGA with a lipid bilayer (DOPC : DSPE-PEG = 0.90 : 0.10) affects its antibacterial activity and cytotoxicity (Figure 3m-n). Using *S. aureus* as a representative for bacteria, we observed that, upon light irradiation ( $0.1 \text{ W/cm}^2$  for 5 min with a solar simulator), the bacterial killing curve (*i.e.*, the relationship between survival *versus* Ce6 dose) of Ce6/PLGA@lipid almost overlapped with that of Ce6/PLGA and, at Ce6 dose  $\geq 4 \mu\text{g/mL}$ , both Ce6/PLGA@lipid and Ce6/PLGA killed 99.99% inoculated *S. aureus* cells, indicative of comparable bactericidal potency, suggesting negligible effects of lipid bilayer coating on the antibacterial potency of free ROS generated by PLGA-encapsulated Ce6 upon light irradiation. On the other hand, using murine macrophage Raw 264.7 cell as a representative for mammalian cell-lines, we found that, upon light irradiation ( $0.1 \text{ W/cm}^2$  for 5 min with a solar simulator), Ce6/PLGA@lipid and Ce6/PLGA resulted in nearly identical relationships of cell survival ratio *versus* Ce6 dose, indicative of comparable cytotoxicity, suggesting negligible effects of lipid bilayer coating on the cytotoxicity of free ROS generated by PLGA-encapsulated Ce6 upon light irradiation.

**Figure 3.** (m) Survival ratios of *S. aureus* cells treated with Ce6/PLGA@lipid (for 10 min, at different Ce6 doses) and then light irradiation (with a solar simulator at  $0.1 \text{ W/m}^2$  for 5 min). The performance of Ce6/PLGA was included for comparison. (n) Survival ratios of Raw 264.7 cells treated with Ce6/PLGA@lipid (for 4 h) and then with light irradiation (with a solar simulator, at  $0.1 \text{ W/m}^2$  for 5-min). The performance of Ce6/PLGA was included for comparison. Data points are reported as mean  $\pm$  standard deviation ( $n = 2$ ).

In addition to Ce6/PLGA, verteporfin pre-loaded PLGA nanoparticle (*i.e.*, Ver/PLGA) was included as a second model for nanoparticles that generate free ROS; verteporfin is a photosensitizer known to generate free  $^1\text{O}_2$  upon irradiation with near-infrared light ( $\lambda \sim 690 \text{ nm}$ ). In similar antibacterial assays and cytotoxicity studies, negligible difference was observed between Ver/PLGA and its lipid bilayer coated counterpart, Ver/PLGA@lipid (Figure S35).

**Figure S35.** (a) Survival ratios of *S. aureus* after 10 min co-incubation with Ver/PLGA or Ver/PLGA@lipid (at different concentrations) and then 5-min irradiation (with a solar simulator at  $0.1 \text{ W/m}^2$ ). Data points are reported as mean  $\pm$  standard deviation. (b) Survival ratios of murine macrophage Raw 264.7 cells after 4-h co-incubation with Ver/PLGA or Ver/PLGA@lipid at different concentrations and then 5-min irradiation (with a solar simulator at  $0.1 \text{ W/m}^2$ ). Data points are reported as mean  $\pm$  standard deviation ( $n = 2$ ).

Collectively, above results suggest that coating a free ROS-generating nanoparticle with a lipid bilayer (*e.g.*, one composed of DOPC : DSPE-PEG =

0.90 : 0.10 as used herein) does not impact the particle's ability to oxidize ROS probes in the bulk solution, its antibacterial activity, or its cytotoxicity to mammalian cells.

If AgPd0.38 generates free ROS as do light-irradiated Ce6/PLGA and Ver/PLGA, coating AgPd0.38 with a same lipid bilayer (DOPC : DSPE-PEG = 0.90 : 0.10) should not impact its antibacterial activity, which is nevertheless not the case. In fact, AgPd0.38 exhibited significantly stronger bactericidal potency than its lipid bilayer coated counterpart, AgPd0.38@lipid (Figure 3l). Indeed, MBC99.9 (the minimal bactericidal concentration to kill 99.9% inoculated bacteria) of AgPd0.38@lipid was 4-fold higher than that of the bare AgPd0.38 and this was the case for all four examined bacterial strains, indicative of significantly weakened bactericidal potency due to the presence of a lipid bilayer coating. A lipid bilayer composed of DOPC : DSPE-PEG = 0.90 : 0.10 is permeable to free 1O2 (as indicated by using Ce6/PLGA and Ver/PLGA as models for free-1O2 generating nanoparticles), and the ROS generated by AgPd0.38 resemble 1O2 in chemical reactivity (Figure 1d-e). Therefore, the distinct effect of lipid bilayer coating on AgPd0.38 *versus* free ROS-generating nanoparticles (*e.g.*, Ce6/PLGA) must arise because the ROS generated by AgPd0.38 is distinct from the freely diffusive ROS produced by light-irradiated Ce6/PLGA. In another word, the ROS generated by AgPd0.38 must be surface-bound.

It should be noted that lipid peroxidation might have occurred on AgPd0.38@lipid, because AgPd0.38@lipid, though 4-fold weaker than AgPd0.38 in killing bacteria, still managed to kill 99.9% inoculated bacterial cells at doses ranging 32-64 μg/mL (Figure 3l). Nevertheless, the resulting lipid-based radicals, though highly oxidative, cannot diffuse freely into the environment due to hydrophobic effects but are confined within the lipid bilayer structure on AgPd0.38, which limits their oxidative reactivity to substances that are in contact with these lipid-based radicals. Supportive evidences to this claim were found in the observations that coating AgPd0.38 with platelet membrane (Figure 3g), which naturally contains lipids to significant extent, failed to make AgPd0.38 more cytotoxic to 4T1 cells (Figure 3i), despite that the platelet membrane coating conferred AgPd0.38 with significantly higher cellular uptake by murine breast cancer 4T1 cells (Figure 3h) (owing to specific recognition between P-selectin naturally present on platelet membrane and CD44 receptors up-regulated on surfaces of cancer cells (*e.g.*, 4T1 cells) (*Nat. Rev. Cancer.* 2011,11, 123-134; *Nature* 2015, 526, 47-48).

**Figure 3.** (g) Schematic illustration on coating AgPd0.38 with a platelet membrane, which yields AgPd0.38@P. There exist specific recognition between P-selectin naturally present in platelet membrane and CD44 receptors over-expressed on murine breast cancer 4T1 cells. (h) Cellular uptake of AgPd0.38@P by 4T1 cells, with that of AgPd0.38 in similar assays included for comparison. \*\*\* indicates  $p < 0.001$ , analyzed by two-sided Student's t test. (i) Survival ratios of 4T1 cells after 4-h treatment with AgPd0.38 or AgPd0.38@P at differing concentrations *in vitro*. Data points are reported as mean  $\pm$  standard deviation ( $n = 2$ ).

5. To confirm ROS not freely dispersing, a transwell experiment or immobilizing AgPd nanocages on a solid surface can be used to check if it can kill bacteria or not. I suspect the conclusion that ROS is only bound on AgPd surface and cannot freely disperse. If it is true, the antibacterial activity will be severely reduced under physiological condition in which many biomolecules and cells may attach on AgPd surface and block the interaction with bacteria. However, according to the *in vivo* experiments, AgPd nanocages still showed considerable bacterial killing. Especially when it is immobilized on catheter, CA/PDA/AgPd still can promote wound healing.

Author Response: We thank the reviewer for the critical comments and inspiring suggestion. In revision, we carried out protein adsorption assays using bovine serum albumin (BSA) as a model protein and found that coating AgPd0.38 to the PDMS wafer surface yielded PDMS/PDA/AgPd surface that, after 5-day co-incubation with BSA in water, exhibited significantly lower protein adsorption than the pristine PDMS and the intermediate PDMS/PDA. Moreover, additional antibacterial assays show that, after 3-h co-incubation, PDMS/PDA/AgPd killed planktonic bacteria in the bulk solution with significantly higher potency than the pristine PDMS and the intermediate PDMS/PDA, owing to the presence of AgPd0.38. Collectively, these results demonstrate that coating AgPd0.38 onto an inert surface converted the surface to be antifouling and antibacterial, which may explain why AgPd0.38-coated substrates remained biofilm inhibitive in complex environments (*e.g.*, complex and protein-rich tryptic soy broth (TSB) used in the

*in vitro* anti-biofilm assays (Figure 4), bacterial infected wounds in mouse models (Figure 5)).

**Figure S39.** (a) Adsorption of bovin serum albumin (BSA) on PDMS/PDA/AgPd after 5-day co-incubation in water. Controls are the pristine PDMS and the intermediate PDMS/PDA. Data points are reported as mean  $\pm$  standard deviation ( $n = 2$ ). (b) Plate-killing assays of PDMS/PDA/AgPd against planktonic bacteria. Controls are the pristine PDMS and the intermediate PDMS/PDA. Data points are reported as mean  $\pm$  standard deviation ( $n = 2$ ).

In fact, the ability to endow an inert surface with the ability to inhibit biofilm formation under complex conditions is not exclusive to AgPd0.38. A prior report shows that coating V2O5 nanoparticle, a haloperoxidase mimicry, onto an inert substrate enables the substrate to thwart biofilm formation even in ocean (*Nat. Nanotechnol.* **7**, 530-535 (2012)). So, we boldly expect that the capacity of enabling an inert surface to inhibit biofilm formation under complex conditions may be a universal property for ROS-generating nanozymes.

We agree with the reviewer that a transwell experiment is useful for detecting whether separation of two agents/materials indeed prevent the potential interactions between them from happening. The reasons why we did not carry out transwell experiments are because the Matrigel commonly used to separate the two to-be-examined agents/materials in a transwell experiment is too thick compared to ROS' effective radii of action (less than a few hundred nanometers). Even if AgPd0.38 failed to oxidize the ROS probes or bacterial cells on the other side of the Matrigel, it would have been difficult for us to say whether it happened because free ROS got extinct on their way of translocating across the Matrigel or whether the presence of the Matrigel prevented the surface-bound ROS from getting in contact with the ROS probes/bacteria.

6. In Figure S17, SYTO-9 can stain both live and dead bacteria.

Author Response: We agree with the reviewer that SYTO-9 stains both live and dead bacteria. The reasons why, in our original manuscript, we stated that “SYTO-9 was used to label live bacteria” are as follows.

To differentiate live and dead bacteria under fluorescence microscopy, we used the Live/Dead BacLight bacterial viability kit provided by Molecular Probes, which provides a two-color fluorescence assay of bacterial viability. According to the product information on the Live/Dead BacLight bacterial viability kit provided by Molecular Probes (<https://www.thermofisher.com/document-connect/document-connect.html?url=https%3A%2F%2Fassets.thermofisher.com%2FTFS-Assets%2FLSG%2Fmanuals%2Fmp07007.pdf&title=TELWRSYjNDc7REVBRCAmbHQ7aSZndDtCYWMmbHQ7L2kmZ3Q7TGlnaHQgOmFjdGVyaWFsIFZpYWJpbGl0eSBLaXRz>), there is a paragraph on how this kit works, which reads:

“The LIVE/DEAD *BacLight* Bacterial Viability Kits utilize mixtures of our SYTO® 9 green-fluorescent nucleic acid stain and the red-fluorescent nucleic acid stain, propidium iodide. These stains differ both in their spectral characteristics and in their ability to penetrate healthy bacterial cells. When used alone, the SYTO 9 stain generally labels all bacteria in a population — those with intact membranes and those with damaged membranes. In contrast, propidium iodide penetrates only bacteria with damaged membranes, causing a reduction in the SYTO 9 stain fluorescence when both dyes are present.”

Based on the information conveyed in this paragraph, we stated in our original manuscript that “SYTO-9 and propidium iodide (PI) were used to label live and dead bacterial cells, respectively”. Still, we have to admit that, even in bacteria with damaged membranes, which are to be stained by propidium iodide, the green fluorescence of SYTO-9, though reduced due to PI’s presence, may still exist, as reminded by the reviewer.

In revision, to more accurately reflect the mechanism by which the two-color fluorescence assay mediated by SYTO-9 and PI works, we have accordingly revised our statement. It now reads:

“... SYTO 9 (green) and PI (red) were used to label all and dead bacterial cells, respectively. ...”

7. The style of references need to be checked carefully.

Author Response: We thank the reviewer for the kind reminder. In revision, we have accordingly checked the references carefully, to ensure that their style meets the requirement by the journal.

### ***Response to Reviewer #2:***

*In this manuscript, Gao et al. synthesized AgPd nanoenzymes for the treatment against bacterial infection, among which AgPd0.38 seems to have the best antibacterial efficacy against different bacteria strains. By generating surface-bound ROS, the nanoenzyme showed preferential killing of bacteria over mammalian cells in vitro and promoted wound disinfection in mouse models. The AgPd0.38 nanoenzyme also showed ability to kill antibiotic-resistant bacteria, inhibit biofilm formation, and suppress immune response in vivo. Overall, the reported nanoenzyme shows promising antimicrobial activity and may have good applications. However, the current version of the manuscript lacks fundamental understandings of the system. There is little knowledge on the preferential killing to bacteria over cells; the proposed mechanism and associated experimental proofs demonstrate inadequate support. Below are some detailed comments that may help the authors to further improve the manuscript.*

*1. The title of the manuscript can be improved to better reflect the content of the manuscript. The authors aimed for a big theme, but only completed a small part. Particularly, neither preferential killing nor surface-binding have been clearly demonstrated throughout the manuscript.*

Author Response: We thank the reviewer for the critical comments. In revision, we have accordingly checked the literature carefully and performed additional experiments using thermally reduced TiO2 nanoparticle, which is known to generate surface-stabilized superoxide radicals in the dark (*J. Phys. Chem. C*, 2007, 111, 10630; *Res. Chem. Intermed.*, 2003, 29, 449–465), as an extra model for nanozymes that generate surface-bound nanoparticle and observed that, in assays performed in the dark, the thermally reduced TiO2 nanoparticle (*i.e.*, R-TiO2) killed *B. subtilis* and *E. coli* (representatives for Gram-positive and -negative bacteria) with appreciable efficiency (Figure 3p-q) but imposed negligible cytotoxicity to mammalian cells (Figure 3r). In contrast to R-TiO2, its untreated precursor TiO2 nanoparticle (*i.e.*, TiO2) was inactive against bacteria under comparable conditions (Figure 3p-q). R-TiO2's observed preferentiality in

eliminating bacteria over mammalian cells was attributed to its ability to generated surface-stability superoxide radical in the dark (*J. Phys. Chem. C*, 2007, 111, 10630; *Res. Chem. Intermed.*, 2003, 29, 449–465) (Figure S36d-e).

**Figure 3.** (p-q) In vitro plate-killing assays of R-TiO2 against (p) *B. subtilis* and (q) *E. coli*. Assays performed similarly but with TiO2 are included for comparison. \* and \*\* indicate  $p < 0.05$  and  $p < 0.01$ , respectively, analyzed by two-sided Student's t test. Data points are reported as mean  $\pm$  standard deviation ( $n = 2$ ). (r) In vitro cell viability assays of R-TiO2 to NIH-3T3, Raw 264.7 and Ana-1 cells. Data points are reported as mean  $\pm$  standard deviation ( $n = 2$ ).

**Figure S36.** (c) Schematic illustration on the oxidation of dihydroethidium (DHE) by superoxide radical. (d-e) Fluorescence emission spectra of DHE after 3-h treatment in the dark with (d) TiO2 and (e) R-TiO2 at different concentration.

Moreover, literature research revealed that FeN5 SA/CNF, a single-atom nanozyme that in principle generates oxygen adatom, efficiently kills a wide-spectrum of bacteria but fails to impact the viability of human colon mucosal epithelial NCM460 cells, thereby leading to effective wound disinfection without negatively affecting the skin tissue (*Sci. Adv.*, 2019, 5, eaav5490).

Inspiringly, our AgPd0.38 nanocage, thermally reduced TiO2 nanoparticle (*i.e.*, R-TiO2), and FeN5 SA/CNF and unanimously exhibited antibacterial potency to appreciable extent but imposed negligible cytotoxicity to mammalian cells even

at much higher doses, despite that they are distinct in material composition and structure and generate drastically differing ROS types ( $^1\text{O}_2$ ,  $\text{O}_2^-$ , and oxygen adatom, respectively). Such a strong similarity in bioactivity despite of their distinction in material encouraged us to propose that preferential killing of bacteria over mammalian cells is a property universal nanozymes capable of generating surface-bound ROS. Therefore, in the revised manuscript, we have accordingly added our results on the performances of thermally reduced  $\text{TiO}_2$  nanoparticle, briefly discussed the prior report on  $\text{FeN}_5$  SA/CNF, and kept the title almost the same as in the original manuscript except that the word “nanoenzymes” therein was changed to “nanozymes” in response to Reviewer #1’s suggestion.

*2. Reduced efficiency from lipid-coated AgPd doesn’t necessarily mean the surface-binding of ROS, which could be from other factors such as fast ROS reacting with lipid, slow diffusion of the substrate, etc. It’s not convincing to attribute the reason of reduced efficiency to the surface-binding without ruling out other possibilities.*

Author Response: We thank the reviewer for the critical comment. Possible reasons why lipid bilayer-coated  $\text{AgPd}_{0.38}$  (*i.e.*,  $\text{AgPd}_{0.38}$ @lipid) exhibited reduced efficiency in oxidizing ROS probes (ascorbic acid (AA), SOSG) in the bulk solution (Figure 11) and in killing bacteria (Figure 31) include (1) hindered outward translocation of the ROS generated by  $\text{AgPd}_{0.38}$  due to its/their surface-bound nature, (2) fast reaction of ROS with the lipid coating over  $\text{AgPd}_{0.38}$ , and (3) reduced ROS production due to slow translocation (*via* diffusion) of  $\text{O}_2$  (the substrate) across the lipid bilayer.

Using Ce6/PLGA as a model for free ROS-generating, we examined how coating it with a lipid bilayer (DOPC : DSPE-PEG = 0.90 : 0.10) affects the ability of the free ROS generated by PLGA-encapsulated Ce6 upon light irradiation to oxidize SOSG, a  $^1\text{O}_2$ -sensitive probe, in the bulk solution around. Our results (Figure 1f-h) show that, after 10 min co-incubation, the as-coated nanoparticle (*i.e.*, Ce6/PLGA@lipid) rendered SOSG to exhibit a fluorescence emission spectrum nearly identical to that of SOSG treated similarly but with the bare Ce6/PLGA particle (Figure 1h), indicative of negligible effects of lipid bilayer coating on the ability of Ce6/PLGA to oxidize SOSG. Similar results were observed when using verteporfin-preloaded PLGA nanoparticle (*i.e.*, Ver/PLGA) as a second model for free ROS-generating nanoparticles (Figure S34); verteporfin is a photosensitizer that generates free  $^1\text{O}_2$  upon light irradiation. Collectively, these results suggest that the lipid bilayer (DOPC : DSPE-PEG = 0.90 : 0.10) is permeable to  $\text{O}_2$  and free  $^1\text{O}_2$  and lacks ability to slow down the influx of  $\text{O}_2$  and

efflux of free  $^1\text{O}_2$ . Combined with the close resemblance in chemical reactivity between the ROS generated on  $\text{AgPd}_{0.38}$  and  $^1\text{O}_2$  (Figure 1d-e and Figure S14-15), the inability of lipid bilayer (DOPC : DSPE-PEG = 0.90 : 0.10) to slow down the influx of  $\text{O}_2$  encouraged us to believe that the ROS production by  $\text{AgPd}_{0.38}$ @lipid (which has the same lipid bilayer as the coating materials) must be at the same efficiency as that on the bare  $\text{AgPd}_{0.38}$ , which excluded the possibility #3 in the list above.

We agree with the reviewer that fast reactions of ROS with lipid molecules in the lipid bilayer coating over  $\text{AgPd}_{0.38}$  may exist, especially considering the significant content of DOPC, a lipid with unsaturated carbon-carbon double bond, in the lipid bilayer (DOPC : DSPE-PEG = 0.90 : 0.10) we used to coat  $\text{AgPd}_{0.38}$ . Such ROS-consuming reactions may result in (a) reduced efficiency of ROS in oxidizing the oxidizable substances (*e.g.*, ROS probes) in the bulk solution due to ROS consumption by lipid bilayer coating, (b) enhanced efficiency of the particle in oxidizing the oxidizable substances in the vicinity around the particle due to lipid radicals generated *via* lipid peroxidation by ROS, or (c) both. The key is to dissect the effects of the consequences above and figure out which is the one that occurs or dominates. Interestingly, using Ce6/PLGA as a model for free ROS-generating nanoparticles, we coated Ce6/PLGA with a same lipid bilayer coating (Figure 1f-g) and found that, upon light irradiation, the resulting Ce6/PLGA@lipid nanoparticle rendered SOSG in the bulk solution to exhibit a fluorescence emission spectrum nearly identical to that of SOSG treated similarly but with the bare Ce6/PLGA particle (Figure 1h), indicative of unaffected outward translocation of the free ROS generated by PLGA-encapsulated Ce6 despite of the presence of the lipid bilayer coating, suggesting that the lipid bilayer coating is permeable to free ROS ( $^1\text{O}_2$  in this specific case). Similar results were observed when using Ver/PLGA as a second model for free ROS-generating nanoparticles (Figure S34). Taken together, above observation suggest that the lipid bilayer coating is permeable free ROS and failed to affect their efficiency to oxidize ROS probes in the bulk solution. In another word, the consumption of ROS by the lipid bilayer coating, even if existing, is to negligible extent, which ruled out the possibility # 2 in the list above.

The distinction between  $\text{AgPd}_{0.38}$  and the free ROS-generating nanoparticles (*e.g.*, Ce6/PLGA, Ver/PLGA) is not limited to the efficiency in oxidizing ROS probes in the bulk solution but also demonstrated in antibacterial assays and mammalian cell viability studies (Figure 3m-n). Using Ce6/PLGA as a model for free ROS-generating nanoparticles, we found that, upon light irradiation ( $0.1 \text{ W/cm}^2$  for 5 min with a solar simulator), Ce6/PLGA@lipid and Ce6/PLGA resulted in almost overlapping relationships of *S. aureus* survival ratio *versus* Ce6 dose and, at Ce6 dose  $\geq 4 \mu\text{g/mL}$ , both Ce6/PLGA@lipid and Ce6/PLGA killed 99.99% inoculated *S. aureus* cells (Figure 3m), indicative of comparable bactericidal potency

between Ce6/PLGA@lipid and Ce6/PLGA, suggesting negligible effects of lipid bilayer coating on the antibacterial potency of free ROS. Moreover, using murine macrophage Raw 264.7 cell as a representative for mammalian cell-lines, we found that, upon light irradiation ( $0.1 \text{ W/cm}^2$  for 5 min with a solar simulator), Ce6/PLGA@lipid and Ce6/PLGA resulted in nearly identical relationships of cell viability *versus* Ce6 dose (Figure 3n), indicative of comparable cytotoxicity, suggesting negligible effects of lipid bilayer coating on the cytotoxicity of free ROS. Similar results were observed by using Ver/PLGA as a second model for free ROS-generating nanoparticles (Figure S35). In stark contrast, coating a same lipid bilayer over AgPd0.38 resulted in 4-fold lower efficiency in killing bacteria (Figure 3l). Taken together, these results indicate that, compared to free ROS, the ROS generated on AgPd0.38 must be distinct in ability to translocate across the lipid bilayer. In another word, the ROS on our AgPd nanocages must be surface-bound.

*3. The authors claimed the endocytosis of AgPd by mammalian cells may confine the cytotoxicity of the ROS within endocytic vesicles, despite only 27% of AgPd could be taken up by the co-incubated cells. Is it possible that mammalian cells are less sensitive to the ROS generated by AgPd than the bacteria? Therefore, no matter where ROS is located, it only has modest damage to the cells.*

Author Response: We thank the reviewer for the critical comments. In revision, we employed two different ways to enhance the cellular uptake of AgPd0.38 by mammalian cells and found that enhanced cellular uptake failed to make AgPd0.38 more cytotoxic (Figure 3h-k). One way we used is to coat AgPd0.38 with platelet membrane (Figure 3g) and, after 4-h co-incubation with murine breast cancer 4T1 cells, the as-coated AgPd0.38@P particle exhibited significantly higher cellular uptake by than the bare AgPd0.38 (~40% *versus* ~5%) (Figure 3h), thanks to the specific recognition between P-selectin naturally present on platelet membrane and CD44 receptor up-regulated on surfaces of cancer cells (*J. Physiol. Cell Physiol.* 2008, 294, C907-C916; *Nat. Rev.Cancer* 2011, 11, 123-134). Nevertheless, AgPd0.38@P failed to reduce the survival ratio of 4T1 cells relatively by ~15% as did the bare AgPd0.38 (Figure 3i), indicative of lack of cytotoxicity despite of significantly enhanced cellular uptake. Another way we tried is to replace adherent cells normally used in cell studies with suspended cells. Specifically, we used the murine macrophage Ana-1 cell as a representative for mammalian cell-lines and found that suspended Ana-1 cells, though offering significantly higher internalization efficiency of AgPd0.38 than adherent Ana-1 cells (~40% *versus* ~25%) (Figure 3j), exhibited almost unimpacted survival (<10% reduction in survival ratio) after 4-h treatment with AgPd0.38 as did the adherent Ana-1 cells, indicative lack of cytotoxicity despite of its absolutely appreciable

efficiency in internalizing AgPd0.38 (Figure 3k). Collectively, these results demonstrate that the relatively low cellular uptake efficiency (~27%) of AgPd0.38 is not the reason why it lacks cytotoxicity to mammalian cells.

**Figure 3.** (h) Cellular uptake of AgPd0.38@P by 4T1 cells, with that of AgPd0.38 in similar assays included for comparison. \*\*\* indicates  $p < 0.001$ , analyzed by two-sided Student's *t* test. (i) Survival ratios of 4T1 cells after 4-h treatment with AgPd0.38 or AgPd0.38@P at differing concentrations *in vitro*. (j) Cellular uptake of AgPd0.38 by adherent *versus* suspended Ana-1 cells. \* indicates  $p < 0.05$ , analyzed by two-sided Student's *t* test. (k) Survival ratios of adherent *versus* suspended Ana-1 cells after 4-h treatment with AgPd0.38 at differing concentrations *in vitro*. Data points are reported as mean  $\pm$  standard deviation ( $n = 2$ ).

To exclude the possibility that AgPd0.38 lacks cytotoxicity because mammalian cells are naturally less sensitive to ROS, we examined the antibacterial activity and cytotoxicity of free ROS-generating nanoparticles and compared their performances with that of AgPd0.38. Using Ce6/PLGA and Ce6/PLGA@lipid as a first set of models for free ROS-generating nanoparticles, we found that, upon light irradiation, both Ce6/PLGA and Ce6/PLGA@lipid killed 99.99% inoculated *S. aureus* cells at Ce6 dose of 4 μg/mL and eliminated ~70% of treated murine macrophage Raw 264.7 cells at Ce6 dose of only 16 μg/mL (Figure 3m-n), despite that neither the light irradiation we used nor the un-irradiated Ce6/PLGA impacted the bacteria or mammalian cells. Similar results were observed by using Ver/PLGA and Ver/PLGA@lipid as our second set of models for free ROS-generating nanoparticles (Figure S34-35); both Ver/PLGA and Ver/PLGA@lipid killed >90% inoculated *S. aureus* cells at 16 μg/mL and eliminated ~70% treated macrophage Raw 264.7 cells at only 4 μg/mL in comparable assays (Figure S35). Collectively, these observations demonstrate that mammalian cells can be effectively eliminated by free ROS, which is in fact the rationale underlying anti-tumor photodynamic therapy. In contrast to these free ROS-generating nanoparticles, AgPd0.38 failed to eliminate 10% of treated mammalian cells (Raw 264.7, NIH-3T3, and 4T1) at doses as high as 100 μg/mL (Figure 2f), despite of its ability to eliminate 99.9% inoculated bacterial cells at only 4-16 μg/mL (varying slightly depending on specific bacterial strain tested) (Figure 2a), suggesting that AgPd0.38 lacks cytotoxicity to mammalian cells. The

ROS generated by AgPd0.38 closely resemble 1O2 in chemical reactivity, and Ce6 and verteporfin are known to produce free 1O2 upon light irradiation. Therefore, we attribute the observed distinction between AgPd0.38 and above free ROS-generating nanoparticles to that the ROS on AgPd0.38 must be surface stabilized, rather than that mammalian cells are naturally less sensitive to ROS.

**Figure 3.** (m) Survival ratios of *S. aureus* cells treated with Ce6/PLGA@lipid (for 10 min, at different Ce6 doses) and then with light irradiation (with solar simulator at  $0.1 \text{ W/m}^2$  for 5-min). Controls are *S. aureus* cells treated similarly but with Ce6/PLGA. (n) Survival ratios of murine macrophage Raw 264.7 cells treated with Ce6/PLGA@lipid (for 4 h) and then with light irradiation (with a solar simulator, at  $0.1 \text{ W/m}^2$  for 5-min). Controls are Raw 264.7 cells treated similarly but with Ce6/PLGA. Data points are reported as mean  $\pm$  standard deviation ( $n = 2$ ).

**Figure S35.** (a) Survival ratios of *S. aureus* after 10 min co-incubation with Ver/PLGA or Ver/PLGA@lipid (at different concentrations) and then 5-min irradiation (with a solar simulator at  $0.1 \text{ W/m}^2$ ). Data points are reported as mean  $\pm$  standard deviation. (b) Survival ratios of murine macrophage Raw 264.7 cells after 4-h co-incubation with Ver/PLGA or Ver/PLGA@lipid at different concentrations and then 5-min irradiation (with a solar simulator at  $0.1 \text{ W/m}^2$ ). Data points are reported as mean  $\pm$  standard deviation ( $n = 2$ ).

4. Higher IC50 in cells than those in bacteria doesn't necessarily mean the preferential killing, again other factors need to be carefully considered. Presumably, a similar nanoenzyme without any endocytosis could present similar effect. The higher sensitivity of bacteria to AgPd doesn't mean the off-target to cells has been reduced (Line 328).

Author Response: We thank the reviewer for the critical comments. Note that it is extremely difficult to find a nanoparticle without any endocytosis, because mammalian cells naturally internalize nanoparticles via endocytosis (though at differing efficiency depending particle and cell-line specifics). Therefore, in revision, we tried to suppress the endocytosis of AgPd0.38 by using endocytosis inhibitors (Figure 3e) and identified the effective endocytosis inhibitors for AgPd0.38. Intriguingly, suppressing the cellular uptake of AgPd0.38 by murine macrophage Raw 264.7 cells with dynasore (Figure 3e), an endocytosis inhibitor, was found to be able to make AgPd0.38 more cytotoxic to these cells (Figure 3f), despite that dynasore itself lacks cytotoxicity under comparable conditions (Figure S31).

In revision, we found that AgPd0.38@P failed to eliminated ~10% of treated 4T1 cells as did the bare AgPd0.38 (Figure 3i), despite that the former exhibited 4-fold higher cellular uptake than the latter (Figure 3h). Similarly, bare AgPd0.38 failed to eliminate 10% of Ana-1 cells no matter whether the cells were adherent or suspended (Figure 3k), despite that the particle was uptaken at 2-fold higher efficiency by suspended Ana-1 cells than by adherent Ana-1 cells (Figure 3j). These results demonstrate that increasing AgPd's cellular uptake cannot make the particle cytotoxic to appreciable extent. In another word, the relatively low cellular uptake efficiency (~27%) of AgPd0.38 is not the reason why AgPd0.38 lacks cytotoxicity to mammalian cells.

In our *in vitro* assays, AgPd0.38 killed 99.9% of inoculated bacteria at 4-16 µg/mL (with slight variation depending specific bacterial strain tested) (Figure 2a) but eliminated <20% of inoculated mammalian cells even at doses up to 100 µg/mL (Figure 2f). When applied topically to bacterial-infected wounds in mouse models, AgPd0.38 effectively promoted wound disinfection and healing while suppressing bacterial infection-relevant immune responses (Figure 2h-m), despite that the bacterial-infected wound microenvironment is much more complex than what is in the *in vitro* assays and simulate the niches where bacteria and host cells co-exist. Collectively, these results suggest that AgPd0.38 is preferentially active against bacteria over mammalian cells both *in vitro* and in mouse models.

5. The authors attempted to highlight the biomedical benefits of preferential killing, however without any appropriate controls, it becomes less convincing. Lipid-coated AgPd could be used as a negative control presumably with reduced bacterial killing, while another potent nanoenzyme with no endocytosis could be used as a control to emphasize the necessity of reduced cell uptake for enhanced biosafety of nanoenzyme.

Author Response: We thank the reviewer for the critical comments. Note that it is extremely difficult to find a nanoparticle without any endocytosis, because mammalian cells naturally internalize nanoparticles *via* endocytosis (though at differing efficiency depending particle and cell-line specifics). Therefore, in revision, we tried to suppress the endocytosis of AgPd0.38 by using endocytosis inhibitors (Figure 3e) and identified the effective endocytosis inhibitors for AgPd0.38. Intriguingly, suppressing the cellular uptake of AgPd0.38 by murine macrophage Raw 264.7 cells with dynasore (Figure 3e), an endocytosis inhibitor, was found to be able to make AgPd0.38 more cytotoxic to these cells (Figure 3f), despite that dynasore itself lacks cytotoxicity under comparable conditions (Figure S31). These observations suggest that endocytosis has unexpectedly protected mammalian cells from AgPd0.38.

In revision, we also examined whether promoting the cellular uptake of AgPd0.38 helps strengthen its cytotoxicity. Specifically, we employed two different ways to enhance the cellular uptake of AgPd0.38 by mammalian cells and found that enhanced cellular uptake failed to make AgPd0.38 more cytotoxic (Figure 3h-k). One way we used is to coat AgPd0.38 with platelet membrane (Figure 3g) and then co-incubate the as-coated AgPd0.38@P particle with murine breast cancer 4T1 cells, because there exists specific recognition between P-selectin naturally present on platelet membrane and CD44 receptor up-regulated on surfaces of cancer cells (*J. Physiol. Cell Physiol.* 2008, 294, C907-C916; *Nat. Rev. Cancer* 2011, 11, 123-134). We found that, after 4-h co-incubation, AgPd0.38@P exhibited significantly higher cellular uptake by than the bare AgPd0.38 (~40% *versus* ~5%) (Figure 3h) but failed to reduce the survival ratio of 4T1 cells relatively by ~15% as did the bare AgPd0.38 (Figure 3i), indicative of lack of cytotoxicity despite of significantly enhanced cellular uptake. Another way we tried is to replace adherent cells normally used in cell studies with suspended cells. Specifically, we used the murine macrophage Ana-1 cell as a representative for mammalian cell-lines and found that suspended Ana-1 cells, though offering significantly higher internalization efficiency of AgPd0.38 than adherent Ana-1 cells (~40% *versus* ~25%) (Figure 3j), exhibited almost unimpacted survival (<10% reduction in survival ratio) after 4-h treatment with AgPd0.38 as did the adherent Ana-1 cells, indicative lack of cytotoxicity despite of its absolutely appreciable efficiency in internalizing AgPd0.38 (Figure 3k). Collectively, these results

demonstrate that the relatively low cellular uptake efficiency ( $\sim 27\%$ ) of  $\text{AgPd}_{0.38}$  is not the reason why it lacks cytotoxicity to mammalian cells.

**Figure 3.** (h) Cellular uptake of  $\text{AgPd}_{0.38}@P$  by 4T1 cells, with that of  $\text{AgPd}_{0.38}$  in similar assays included for comparison. \*\*\* indicates  $p < 0.001$ , analyzed by two-sided Student's t test. (i) Survival ratios of 4T1 cells after 4-h treatment with  $\text{AgPd}_{0.38}$  or  $\text{AgPd}_{0.38}@P$  at differing concentrations *in vitro*. (j) Cellular uptake of  $\text{AgPd}_{0.38}$  by adherent *versus* suspended Ana-1 cells. \* indicates  $p < 0.05$ , analyzed by two-sided Student's t test. (k) Survival ratios of adherent *versus* suspended Ana-1 cells after 4-h treatment with  $\text{AgPd}_{0.38}$  at differing concentrations *in vitro*. Data points are reported as mean  $\pm$  standard deviation ( $n = 2$ ).

6. Besides plots, other experimental methods are much anticipated, such as microscopic imaging, protein blots, ROS sensing kinetics, etc. Moreover, a schematic illustration could better support the explanations.

Author Response: We thank the reviewer for the critical comments and kind suggestion. In revision, we prepared a few schematic illustrations and also carried out more microscopy characterizations (*e.g.*, synchrotron soft X-ray microscopy imaging), to help us better illustrate the work.

\*\* See Nature Research's author and referees' website at [www.nature.com/authors](http://www.nature.com/authors) for information about policies, services and author benefits.

## **REVIEWERS' COMMENTS**

### **Reviewer #1 (Remarks to the Author):**

The authors have addressed all the concerns of the reviewers and improved the work with more detailed analyses. I think it is acceptable for publication in current version.

### **Reviewer #2 (Remarks to the Author):**

Overall, the authors have done impressive works in revising the manuscript, which reads much stronger and the conclusion is more convincing now. However, there are still two minor comments to for the authors to consider:

1. The title is still somewhat awkward. Its readability and accuracy should be improved.
2. The fundamental understandings and knowledge drawn from the newly added experiments should be emphasized in the conclusion section.

## ***Responses to Reviewer Comments***

The nanozymes that preferentially kill bacteria over mammalian cells and bio-compatibly inhibit biofilm formation

*Feng Gao#, Tianyi Shao#, Yunpeng Yu, Yujie Xiong\*, and Lihua Yang\**

~~~~~

Response to Reviewer 1:

The authors have addressed all the concerns of the reviewers and improved the work with more detailed analyses. I think it is acceptable for publication in current version.

Author Response: We gratefully thank the reviewer for the positive remarks.

Response to Reviewer 2:

Overall, the authors have done impressive works in revising the manuscript, which reads much stronger and the conclusion is more convincing now.

Author Response: We gratefully thank the reviewer for the positive remarks.

*However, there are still two minor comments to for the authors to consider:
1. The title is still somewhat awkward. Its readability and accuracy should be improved.*

Author Response: We gratefully thank the reviewer for the critical comment. In revision, we have accordingly revised the title. Now, the title reads:

“The nanozymes that preferentially kill bacteria over mammalian cells and bio-compatibly inhibit biofilm formation”

With this change, we hope that the current title could offer improved readability and accuracy.

2. The fundamental understandings and knowledge drawn from the newly added experiments should be emphasized in the conclusion section.

Author Response: We gratefully thank the reviewer for the inspiring suggestion. In revision, we have accordingly incorporated the fundamental understanding and knowledge drawn from the newly added experiments into the conclusion section. Now, the conclusion section reads:

“In summary, we demonstrate the feasibility of using nanozymes that generate surface-bound ROS to kill selectively bacteria over mammalian cells. The result was derived from analysis on three distinct nanozymes that universally produce surface-bound ROS, with an oxidase-like silver-palladium bimetallic alloy nanocage, AgPd_{0.38}, being the lead one. The selectivity is attributable to the surface-bound nature of the as-generated ROS and an unexpected antidote role of endocytosis, a cellular process that is common to mammalian cells but absent in bacteria; in fact, nanoparticles that generate diffusive ROS are toxic indiscriminately to bacteria and mammalian cells, and inhibiting—rather than facilitating—endocytosis sensitizes mammalian cells to AgPd_{0.38}. Though generating surface-bound ROS, AgPd_{0.38} efficiently eliminated antibiotic-resistant bacteria and effectively delayed the onset of bacterial resistance emergence. When used as a coating additive, AgPd_{0.38} enabled an originally inert surface to inhibit biofilm formation and suppress the infection-related immune responses in mouse models. This work opens an avenue toward biocompatible nanozymes and may have implications in our fight against both genetically-encoded and phenotypic AMR.”